# FuseBatch: Unlocking the Potential of Diffusion Models in Throughput Perspective

## Abstract

Diffusion models achieve state-of-the-art image quality but suffer from slow, iterative denoising. Existing acceleration methods focus on reducing the number of iterations, but these approaches are nearing practical limits. To address this, we take a different perspective by improving efficiency through generating multiple images within a single forward pass. We propose **FuseBatch** which fuses multiple inputs into a shared latent, applies the denoiser once, and unfuses the results to recover all outputs. To extend across domains, we introduce **FB-UNet** for pixel-space models and **FB-AE** for latent diffusion models. We further propose **Timestep-Fusion Scheduling (TFS)**, an inference-only strategy that balances throughput and quality, enabling FuseBatch to surpass the baseline at comparable throughput settings. Across DDPM and step-reduction methods (e.g DDIM, Flow Matching), we achieve near-multiplicative throughput gains with modest quality trade-offs, demonstrating its compatibility with existing acceleration techniques. Moreover, it scales effectively to high-resolution LDMs where larger fusion factors become attainable, providing a practical and orthogonal path to faster diffusion sampling.

## 1 Introduction

Diffusion Models (DMs) have demonstrated state-of-the-art performance in image synthesis, delivering high fidelity and diversity across a range of datasets. More recently, latent diffusion models (LDMs) have further advanced this capability, enabling high-resolution image generation with improved efficiency. However, this efficiency comes at the expense of high inference cost. Unlike VAEs and GANs that generate samples in a single step, DMs require iterative denoising over many timesteps, which severely limits throughput[1]. Consequently, recent research has focused on improving the generation throughput of diffusion models (Liu et al., 2022; Lu et al., 2022a).

Despite these efforts, recent approaches are increasingly facing practical limits. Most methods aim to increase throughput by reducing per-sample inference time, primarily through cutting down the number of denoising iterations (Lu et al., 2022b; Zhang & Chen, 2022). For example, DDIM (Song et al., 2020a) reformulates the reverse diffusion process into a non-Markovian, deterministic sampler that can generate high-fidelity images in as few as ten steps by reusing noise predictions across iterations. Flow matching methods (Lipman et al., 2022), on the other hand, learn continuous-time probability flows, enabling direct mappings from noise to data with far fewer function evaluations. However, as the number of iterations becomes smaller, further reductions become increasingly difficult and the throughput gains also diminish. This suggests that throughput is approaching a practical limit and cannot be improved indefinitely by step-reduction alone. To overcome this bottleneck, we depart from reducing inference time per sample and instead open a novel dimension of efficiency by increasing the number of images generated per forward pass.

Building on this motivation, we propose FuseBatch, a framework that processes multiple inputs jointly within a single forward pass and produces multiple outputs simultaneously. Achieving such efficiency is far from straightforward, since it requires solving three interdependent problems: compressing multiple inputs into a representation comparable in scale to a single sample, ensuring that the

---

[1] $\text{Throughput} = \dfrac{\text{images generated}}{\text{total inference time}} = \dfrac{\text{images per forward pass}}{\text{inference time (per forward pass)}}$

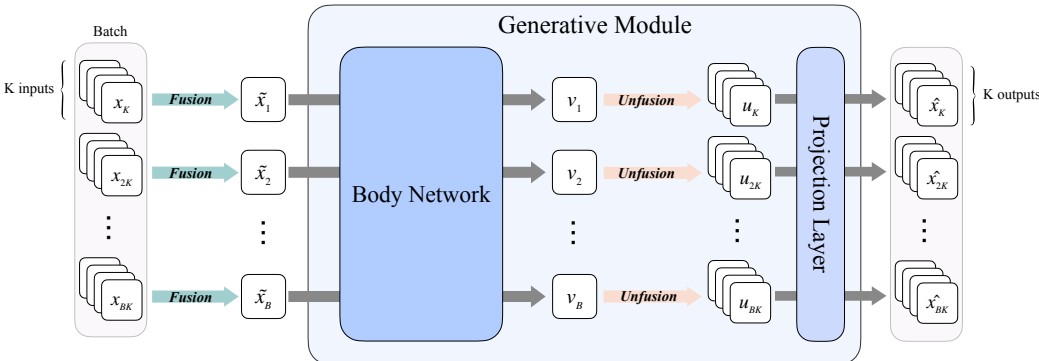

Figure 1: Illustration of the FuseBatch framework. A batch of $K \times B$ inputs is fused into $B$ representations, jointly processed by a shared denoising backbone, then unfused via the *unfusion* module into $K$ representations per group, and finally passed through a projection layer to produce the $K$ outputs. This design enables up to $K\times$ throughput gains with minimal quality loss, and applies broadly across DDPM, DDIM, and flow matching.

fused representation can be meaningfully processed by the model, and then disentangling it to accurately recover each original image. These obstacles demand a dedicated architectural solution rather than simple approaches such as concatenation, which inevitably lose semantic information. To this end, we introduce a lightweight dual-space fusion–unfusion module that constructs a shared latent representation, routes it through the model in a single computation, and restores fine-grained outputs for all inputs (Fig. 1). By elevating throughput to a primary design objective, FuseBatch surpasses the practical limits of step-reduction and establishes a new direction for speeding up diffusion inference.

For completeness, we validate **FuseBatch** across diverse settings to highlight both generality and scalability. We design two architectural variants, each tailored to the dominant bottlenecks of diffusion models in pixel and latent spaces. For pixel-space models such as DDPM, where the UNet dominates computation, we propose **FB-UNet**, which enables joint denoising of multiple inputs even under conditional generation. To further balance throughput and fidelity, we introduce **Temporal Fusion Scheduling (TFS)**, a simple yet effective strategy that dynamically varies the fusion factor $K$ across the denoising trajectory. Both FB-UNet and TFS are fully compatible with step-reduction techniques such as DDIM and flow matching, allowing throughput gains from multi-sample fusion to compound with reductions in denoising iterations. In high-throughput regimes, this combined design surpasses the practical limits of step-reduction alone, achieving levels of performance unattainable by existing methods.

For latent-space models such as LDMs, we propose **FB-AE**. A naïve application of FB-UNet fuses only the denoiser while leaving the autoencoder untouched, limiting the achievable throughput. FB-AE overcomes this by embedding fusion–unfusion directly around the autoencoder, enabling end-to-end acceleration in high-resolution settings. Since latent representations are both compressed and structurally sparse, they are particularly well-suited for multi-sample fusion with minimal information loss. At higher resolutions, the relative degradation from larger fusion scales is further reduced, making the benefits of fusion pronounced. Altogether, FuseBatch extends seamlessly from pixel to latent diffusion models, providing a unified and scalable framework for efficient image synthesis.

## 2 RELATED WORK

Diffusion models have emerged as powerful generative frameworks, achieving state-of-the-art fidelity and diversity by learning to iteratively denoise a corrupted signal (Sohl-Dickstein et al., 2015; Ho et al., 2020; Song & Ermon, 2019). Pixel-space diffusion models such as DDPM operate on raw images, progressively refining noisy inputs into high-quality samples (Ho et al., 2020). To enable higher-resolution generation with greater efficiency, Latent Diffusion Models (LDMs) (Rombach et al., 2022) introduced an autoencoding formulation that compresses images into a latent space before denoising, which has since become the foundation of many large-scale generative models. Both pixel-

and latent-space diffusion models demonstrate strong generative fidelity, but they share a fundamental drawback: inference remains slow due to the need for iterative denoising across timesteps.

To address this limitation, a major line of work has focused on reducing the number of sampling steps for accelerating diffusion. Deterministic samplers such as DDIM reparameterize the reverse process into non-Markovian updates, producing high-quality images in a few(e.g 50) iterations (Song et al., 2020a; Nichol & Dhariwal, 2021). Flow-matching frameworks (Lipman et al., 2022; 2024) extend this idea by learning continuous-time vector fields that directly transport noise to data. More recently, step-aware approaches have been proposed that adapt network capacity across timesteps, though they still operate on a per-sample basis rather than multiplexing multiple inputs (Yang et al., 2023). Despite these advances, as the step budget enters the few-step regime, further reductions increasingly degrade sample quality (Noroozi et al., 2024; Xia et al., 2023; Yue et al., 2023).

There has also been a line of work exploring acceleration superposing inputs into a joint latent trajectory. Murahari et al. (Murahari et al., 2022) subsequently extended input superposition to sentence- and token-level classification, while Menet et al. (Menet et al., 2023) applied the concept to small-scale image datasets such as MNIST and CIFAR. In contrast to these approaches, we are the first attempt to increase throughput by multiplexing multiple signals into a shared latent trajectory on diffusion models for image generation, enabling multiple images to be fused and generated within a single inference while preserving the fidelity required for high-quality synthesis.

## 3 METHOD

Our goal is to *increase throughput not by reducing per-sample latency, but by generating multiple images within a single forward pass*. To achieve this, we propose a dual-space pipeline that fuses $K$ inputs into a single representation, processes them jointly, and recovers $K$ outputs through lightweight heads. We formalize this mechanism by introducing fusion–unfusion operators with index conditioning to preserve sample identity (Sec.3.1). Building on this formulation, we design **FB-UNet**, which reduces the standard $B \times K \times T$ forward passes to $B \times T$ by inserting fusion and unfusion modules around the UNet backbone (Sec.3.2). To further improve the efficiency–quality trade-off, we propose **Timestep-Fusion Scheduling (TFS)**, which adapts the fusion factor $K$ to the relative importance of each timestep in the denoising process(Sec.3.3). Finally, for latent-space models such as LDMs, we extend fusion to the autoencoder via **FB-AE**, enabling shared encoding/decoding and denoising across multiple samples (Sec.3.4).

### 3.1 FORMULATION ANALYSIS

Achieving throughput gains by processing multiple inputs jointly is non-trivial. It requires a mechanism that can (i) compress several inputs into a single latent of consistent scale, (ii) propagate this fused representation through the backbone without loss of compatibility, and (iii) disentangle it to faithfully recover each output. To meet these requirements, we introduce the *dual space fusion–unfusion pipeline*. At a high level, the **fusion** stage condenses $K$ inputs into a single latent representation that retains the essential information of all elements, while the **unfusion** stage recovers the corresponding $K$ outputs by re-injecting sample-specific identity. This mechanism allows the backbone to operate at the dual space of the original model for the fused representation, amortizing computation across multiple samples.

Formally, given a set of $K$ inputs $\{x_i\}_{i=1}^{K}$, a fusion mapper $\phi : (\mathbb{R}^d)^K \to \mathbb{R}^h$ aggregates them into a single latent representation. This fused vector is processed once by a shared network $\tilde{f} : \mathbb{R}^h \to \mathbb{R}^h$, and finally an unfusion module $\phi^{-1}$ decomposes the output back into individual predictions:

$$\{f(x_1), f(x_2), \ldots, f(x_K)\} = \phi^{-1}\Big(\tilde{f}\big(\phi(x_1, x_2, \ldots, x_K)\big)\Big). \tag{1}$$

Whereas the conventional element-wise approach applies $f$ to each input independently, our pipeline introduces a dual operator $\tilde{f}$ that acts once on the fused representation, after which all $K$ outputs are

recovered via $\phi^{-1}$. The formulation is illustrated below:

$$
\begin{array}{ccc}
\phi(x_1, x_2, \ldots, x_K) & \xrightarrow{\ \tilde{f}\ } & \tilde{f}(\phi(x_1, x_2, \ldots, x_K)) \\
\Big\uparrow \phi \ :\text{fusion} & & \Big\downarrow \phi^{-1} \ :\text{unfusion} \\
\{x_1, x_2, \ldots, x_K\} & \xrightarrow{\ f\ } & \{y_1, y_2, \ldots, y_K\}
\end{array}
$$

The fused representation preserves the essential information of all inputs, as latent spaces often exhibit the property that additive operations retain semantic structure rather than destroying it (Zhang et al., 2017; Verma et al., 2019). Furthermore, the unfusion process can recover the original image information from the fused representation by exploiting the distinct high-dimensional signatures of each input (Gandelsman et al., 2019; Szeliski et al., 2000).

For this dual-space pipeline to function correctly, each input $x_j$ must map to its corresponding output $y_j$, preserving bijective correspondence. This precludes a purely permutation-invariant fusion, as it would erase sample identity. Accordingly, the fusion-unfusion stage must maintain index-specific identity across all $K$ elements. To achieve this, we introduce an index-conditioned composition function $\psi$ that injects ordering information into the fused representation. The complete process is thus given by:

$$
\{y_1, y_2, \ldots, y_K\} = \phi^{-1}\Big(\tilde{f}\Big(\phi\big(\psi(x_1, 1), \psi(x_2, 2), \ldots, \psi(x_K, K)\big)\Big)\Big), \tag{2}
$$

where $\psi(x_j, j)$ denotes the index-conditioned encoding of $j$-th sample. This ensures that each fused input can be faithfully disentangled into its corresponding output, enabling information-preserving fusion and unfusion.

## 3.2 FB-UNET

For pixel-space diffusion models such as DDPM, the denoising U-Net constitutes the main computational bottleneck, as every noisy sample must be processed independently at each timestep. Concretely, a standard denoiser $\epsilon_\theta$ requires $B \times K \times T$ forward passes for a batch of size $B$, fusion factor $K$, and sampling horizon $T$. To overcome this inefficiency, we propose **FB-UNet**, a U-Net backbone augmented with the FuseBatch strategy that restructures the computation into $B \times T$ passes. FB-UNet achieves this by inserting two lightweight, learnable modules into the U-Net: a fusion module $F_\phi$ that condenses each group of $K$ noisy inputs into a single latent representation, and an unfusion module $U_{\phi^{-1}}$ that disentangles the fused features back into $K$ individual predictions. Between these modules, the shared denoiser body $\epsilon_\theta^{\text{body}}$ is executed only once, and the recovered features are finally mapped to noise estimates by the projection head $\epsilon_\theta^{\text{proj}}$:

$$
\mathbb{R}^{B \times K \times d} \xrightarrow{F_\phi} \mathbb{R}^{B \times h} \xrightarrow{\epsilon_\theta^{\text{body}}} \mathbb{R}^{B \times h} \xrightarrow{U_{\phi^{-1}}} \mathbb{R}^{B \times K \times h} \xrightarrow{\epsilon_\theta^{\text{proj}}} \mathbb{R}^{B \times d}. \tag{3}
$$

**Fusion via Index-Conditioned Composition** We aggregate each group of $K$ noisy samples $\{x_t^{(i,j)}\}_{j=1}^K$ into a single latent representation $\tilde{x}_t^{(i)} \in \mathbb{R}^h$ using an index-conditioned composition function $C_\psi : \mathbb{R}^{B \times K \times d} \to \mathbb{R}^{B \times K \times h}$ and a fusion function $F_\phi : \mathbb{R}^{B \times K \times h} \to \mathbb{R}^{B \times h}$. For framework-agnostic implementation, we adopt a *parameter-free summation* for $F_\phi$, avoiding the need for learnable parameters $\phi$. However, naïvely summing inputs collapses sample identity, making it unsuitable for recovering individual trajectories. To address this, we first pass each input through a learnable module $C_\psi$ that incorporates the sample index $j$, enabling permutation-sensitive fusion. The fused representation is thus given by:

$$
\tilde{x}_t^{(i)} = F_\phi\left(\{x_t^{(i,j)}\}_{j=1}^K\right) = \sum_{j=1}^K C_\psi\left(x_t^{(i,j)}, j\right) \in \mathbb{R}^h, \quad i \in [B], j \in [K]. \tag{4}
$$

Here, each $C_\psi$ is implemented as a small 2D convolutional block with kernel size 3, using a learnable index-conditioned function $\psi(x_j, j)$ to retain sample identity while preserving spatial structure.

**Shared Denoising Step** To amortize computation, we decompose the original denoiser $\epsilon_\theta$ into a heavy body $\epsilon_\theta^{\text{body}}$ and a lightweight projection head $\epsilon_\theta^{\text{proj}}$ to apply $\epsilon_\theta^{\text{body}}$ exactly once to the fused representation batch, dramatically cutting computation. Concretely, for each batch index $i$, the fused latent $\tilde{x}_t^{(i)}$ and timestep $t$ are fed into $\epsilon_\theta^{\text{body}}$, producing

$$\tilde{v}_t^{(i)} = \epsilon_\theta^{\text{body}}\big(\tilde{x}_t^{(i)}, t\big) \in \mathbb{R}^h. \tag{5}$$

Importantly, this design allows the expensive core to be operated on $\tilde{x}_t$ exactly once per group while preserving the expressive power of the original model. Therefore, by inserting lightweight fusion and unfusion modules around the expensive core with negligible parameter overheads, FuseBatch generates $B \times K$ images with only $B \times T$ full U-Net body evaluations, delivering substantial throughput gains at the expense of subtle reductions in image quality.

**Index-Conditioned Unfusion** We recover each of the $K$ denoising outputs via an efficient, index-conditioned channel-wise disentanglement and projection head. First, we apply $U_{\phi^{-1}}$, a pointwise 2D convolution conditioned on the original sample index $j$. We focus on the fact that the shared pass (Eq. 5) captures rich multi-sample features while leveraging the full spatial capacity of the U-Net backbone. Because $\tilde{v}_t$ resides just before the final projection layer, it already encodes detailed localization, context, and inter-sample correlations (Ronneberger et al., 2015). Therefore, the fused features in $\tilde{v}_t$ are disentangled by applying linear projections along the channel dimension, enabling separation through a lightweight $U_{\phi^{-1}}$ that is invariant to spatial position. Next, each $u_t^{(i,j)}$ is passed through the projection layer to yield the noise prediction:

$$\hat{\epsilon}_t^{(i,j)} = \epsilon_\theta^{\text{proj}}(u_t^{(i,j)}) = \epsilon_\theta^{\text{proj}}(U_{\phi^{-1}}\big(\tilde{v}_t^{(i)}, j\big)), \quad i \in [B], j \in [K]. \tag{6}$$

Finally, we update each sample using the standard diffusion step to produce $x_{t-1}^{(i,j)}$.

## 3.3 Timestep-Fusion Scheduling (TFS)

To achieve a favorable speed–quality tradeoff, we introduce **Timestep-Fusion Scheduling (TFS)**, which dynamically varies the fusion factor $K$ across the denoising trajectory. It is well established that diffusion processes contain timesteps of varying importance, with certain steps being more critical to sample fidelity than others (Nichol & Dhariwal, 2021; Song et al., 2020b; Karras et al., 2022; Yang et al., 2023). Accordingly, TFS allocates smaller fusion factors $K$ to fidelity-critical steps and larger $K$ to less sensitive regions. This design both improves throughput and preserves quality by adapting fusion strength to the intrinsic importance of each timestep, without additional training. For example, in the case of DDPM, early steps dominated by high noise primarily recover low-frequency structure and are relatively robust to aggressive fusion. In contrast, later steps refine high-frequency details, where large $K$ may cause blurring. Accordingly, our DDPM-specific TFS schedule places larger fusion factors in the early phase and gradually reduces them as denoising progresses.

## 3.4 FB-AutoEncoder

**Architecture and Training** We extend the FuseBatch paradigm to latent diffusion models, where applying FB-UNet alone fuses the denoiser but leaves the decoder to process each sample independently, limiting throughput gains. To address this bottleneck, we propose FB-AutoEncoder (FB-AE), which places fusion and unfusion modules around the autoencoder so that both encoding and decoding can be shared across multiple inputs. FB-AE applies the index-conditioned fusion module $F_\phi$ immediately before the encoder $\mathcal{E}_\alpha$ and places the unfusion module $U_{\phi^{-1}}$ just ahead of the decoder's final projection layer $\mathcal{D}_\beta^{\text{proj}}$. To clarify notation, we denote the decoder as $\mathcal{D}_\beta = (\mathcal{D}_\beta^{\text{body}}, \mathcal{D}_\beta^{\text{proj}})$, where $\mathcal{D}_\beta^{\text{body}}$ represents the intermediate convolutional layers that transform latent features and $\mathcal{D}_\beta^{\text{proj}}$ is the final linear projection back to pixel space. By reusing the lightweight convolutional blocks introduced in Sec. 3.2, FB-AE adds only negligible parameter overhead while preserving the original latent-space expressivity. During training, FB-AE learns to map groups of $K$ noisy inputs into a single fused representation, encode and decode them jointly, and then recover $K$ outputs in parallel, as formalized below:

$$\mathbb{R}^{B \times K \times d} \xrightarrow{F_\phi} \mathbb{R}^{B \times h} \xrightarrow{\mathcal{E}_\alpha} \mathbb{R}^{B \times m} \xrightarrow{\mathcal{D}_\beta^{\text{body}}} \mathbb{R}^{B \times m} \xrightarrow{U_{\phi^{-1}}} \mathbb{R}^{B \times K \times h} \xrightarrow{\mathcal{D}_\beta^{\text{proj}}} \mathbb{R}^{B \times K \times d}. \tag{7}$$

Table 1: Unified comparison of Vanilla, FuseBatch ($K = 2, 4$), and TFS across three sampling frameworks (DDPM, DDIM (T=50), FM) on CIFAR-10 and CelebA. Reported metrics include throughput (images/sec), MAC reduction, parameter overhead, and sample quality (FID↓, sFID↓, IS↑, Precision↑, Recall↑). Best results are in **bold**, and second-best results are underlined.

| | CIFAR-10 | | | | | | | | |
|---|---|---|---|---|---|---|---|---|---|
| Method | Variant | Throughput | MAC | Parameter | FID↓ | sFID↓ | IS↑ | Precision↑ | Recall↑ |
| DDPM | Vanilla | 1.71 | 6.07T | 35.7M | **3.41** | 4.40 | 9.20 | **0.68** | 0.58 |
| | FuseBatch (K=2) | 3.17 (×1.85) | 3.07T (50.6%) | +63.1K (0.18%) | 5.93 | 4.58 | 8.67 | 0.63 | 0.58 |
| | FuseBatch (K=4) | **6.31** (×3.49) | **1.55T** (25.5%) | +102.0K (0.29%) | 18.27 | 6.54 | 7.41 | 0.56 | 0.55 |
| | TFS | 3.31 (×1.94) | 3.06T (50.4%) | – | 3.88 | **4.27** | **9.29** | 0.66 | **0.60** |
| DDIM (T=50) | Vanilla | 35.47 | 303.44G | 35.7M | 7.99 | 6.66 | **8.71** | **0.64** | 0.55 |
| | FuseBatch (K=2) | 65.39 (×1.84) | 153.44G (50.6%) | +63.1K (0.18%) | 13.34 | 7.16 | 8.11 | 0.59 | 0.56 |
| | FuseBatch (K=4) | **120.95** (×3.41) | **77.31G** (25.5%) | +102.0K (0.29%) | 26.92 | 7.95 | 7.16 | 0.53 | 0.54 |
| | TFS | 66.80 (×1.88) | 152.87G (50.4%) | – | **10.35** | **6.29** | 8.58 | 0.60 | **0.57** |
| FM | Vanilla | 9.60 | 124.55G | 55.7M | **2.74** | **3.84** | 9.47 | **0.65** | **0.63** |
| | FuseBatch (K=2) | 18.22 (×1.90) | 62.89G (50.5%) | +185.9K (0.33%) | 6.72 | 4.50 | 8.76 | 0.62 | 0.60 |
| | FuseBatch (K=4) | **32.98** (×3.44) | **31.69G** (25.4%) | +323.4K (0.58%) | 21.11 | 7.10 | 7.35 | 0.58 | 0.53 |
| | TFS | 18.29 (×1.91) | 62.70G (50.3%) | – | 4.10 | 3.95 | 8.72 | 0.61 | 0.59 |
| | **CelebA** | | | | | | | | |
| Method | Variant | Throughput | MAC | Parameter | FID↓ | sFID↓ | IS↑ | Precision↑ | Recall↑ |
| DDPM | Vanilla | 0.71 | 15.56T | 113.7M | **3.51** | 6.27 | **2.77** | **0.76** | 0.50 |
| | FuseBatch (K=2) | 1.28 (×1.80) | 7.92T (50.9%) | +63.1K (0.06%) | 4.73 | 6.98 | 2.70 | 0.75 | 0.46 |
| | FuseBatch (K=4) | **2.34** (×3.30) | **4.01T** (25.8%) | +102.0K (0.09%) | 9.15 | 10.34 | 2.64 | 0.70 | 0.40 |
| | TFS | 1.33 (×1.87) | 7.87T (50.6%) | – | 3.81 | **6.00** | **2.77** | 0.74 | **0.51** |
| DDIM (T=50) | Vanilla | 14.56 | 778.00G | 113.7M | 5.58 | 8.37 | 2.81 | **0.74** | 0.46 |
| | FuseBatch (K=2) | 26.05 (×1.79) | 395.88G (51.1%) | +63.1K (0.06%) | 7.43 | 9.07 | 2.75 | 0.71 | 0.42 |
| | FuseBatch (K=4) | 47.67 (×3.27) | 200.32G (25.8%) | +102.0K (0.09%) | 12.62 | 12.26 | 2.67 | 0.65 | 0.36 |
| | TFS | 26.83 (×1.84) | 393.63G (50.6%) | – | **6.52** | **7.82** | **2.84** | 0.68 | **0.48** |
| FM | Vanilla | 2.12 | 589.55G | 169.1M | **2.80** | 5.35 | **3.23** | 0.67 | **0.60** |
| | FuseBatch (K=2) | 4.06 (×1.92) | 297.24G (50.4%) | +185.9K (0.11%) | 3.67 | 5.67 | 3.17 | **0.68** | 0.56 |
| | FuseBatch (K=4) | **7.65** (×3.61) | **149.59G** (25.4%) | +323.4K (0.19%) | 7.15 | 7.24 | 3.00 | 0.62 | 0.52 |
| | TFS | 4.08 (×1.92) | 296.49G (50.3%) | – | 2.98 | **5.20** | 3.22 | 0.66 | 0.59 |

**Inference**   At test time, we skip the initial fusion and encoder. Instead we draw one latent $\tilde{z}_T^{(i)} \sim \mathcal{N}(0, I) \in \mathbb{R}^h$ per group, implicitly encoding the identities of $K$ samples. We then apply the diffusion denoiser $\epsilon_\theta$ over $T$ timesteps to transform $\tilde{z}_T^{(i)}$ into $\tilde{z}_0^{(i)}$. Finally, the decoder body processes fused latent once, and the unfusion module followed by the projection head recover all $K$ images:

$$\tilde{z}_T^{(i)} \xrightarrow[t=T \to 1]{\epsilon_\theta} \tilde{z}_0^{(i)} \xrightarrow{\mathcal{D}_\beta^{\mathrm{body}}} \tilde{v}^{(i)} \xrightarrow{U_{\phi^{-1}}(\cdot, j)} u^{(i,j)} \xrightarrow{\mathcal{D}_\beta^{\mathrm{proj}}} \hat{x}^{(i,j)}, \quad i \in [B], j \in [K]. \tag{8}$$

This procedure achieves upto $K$-fold throughput boost with only modest fusion overhead, generating $B \times K$ images from $B$ random latents, though with noticable sample fidelity trade-offs.

## 4 RESULTS

We evaluate **FuseBatch** across standard DDPMs, class-conditional variants, step-reduced samplers such as DDIM and Flow Matching, and latent diffusion models (LDMs). In standard pixel-space models, including conditional settings, small fusion factors (e.g., $K = 2$) bring near-multiplicative throughput gains with only minor quality loss. However, larger $K$ values, cause suboptimal trade-offs. To address this, we propose **Timestep-Fusion Scheduling (TFS)**, which alters $K$ adaptively across timesteps in the denoising trajectory. This preserves fidelity while sustaining acceleration, and in some cases improves IS, Recall, or sFID. Extending FuseBatch to step-reduction methods demonstrates their orthogonality, as the efficiency gains compound when the two are combined, while TFS helps restore the quality–efficiency balance at larger fusion factors. FuseBatch also surpasses the practical throughput ceilings of step-reduction alone, maintaining quality where conventional methods collapse. Finally, results on LDMs confirm that FuseBatch scales to high-resolution models, where larger fusion factors are increasingly viable due to lower per-pixel information density. Overall, FuseBatch emerges as a general and scalable strategy that boosts throughput while keeping quality competitive.

**Implementation Details**   To underscore the plug-and-play nature of **FuseBatch**, we introduce only the learnable Fusion and Unfusion modules into each base model, *leaving all architectural*

| CIFAR-10 | | | | | | |
|---|---|---|---|---|---|---|
| Throughput | Variant | FID↓ | sFID↓ | IS↑ | Prec.↑ | Rec.↑ |
| 66.80 img/s | Vanilla | 10.36 | 8.16 | 8.54 | **0.63** | 0.53 |
| | TFS | **10.35** | **6.29** | **8.58** | 0.60 | **0.57** |
| 156.56 img/s | Vanilla | 17.55 | 12.66 | 8.15 | **0.59** | 0.46 |
| | TFS | **14.30** | **8.76** | **8.32** | 0.58 | **0.53** |
| 317.89 img/s | Vanilla | 43.59 | 26.38 | 6.83 | 0.50 | 0.30 |
| | TFS | **22.99** | **13.76** | **7.84** | **0.55** | **0.46** |
| 565.03 img/s | Vanilla | 99.09 | 51.83 | 4.80 | 0.38 | 0.15 |
| | TFS | **50.07** | **28.11** | **6.48** | **0.48** | **0.31** |

| CelebA | | | | | | |
|---|---|---|---|---|---|---|
| Throughput | Variant | FID↓ | sFID↓ | IS↑ | Prec.↑ | Rec.↑ |
| 26.83 img/s | Vanilla | 7.12 | 10.68 | 2.77 | **0.73** | 0.43 |
| | TFS | **6.52** | **7.82** | **2.84** | 0.68 | **0.48** |
| 66.90 img/s | Vanilla | 12.33 | 19.01 | 2.59 | **0.73** | 0.30 |
| | TFS | **8.90** | 11.53 | **2.72** | 0.69 | **0.40** |
| 127.92 img/s | Vanilla | 23.06 | 31.96 | 2.37 | 0.68 | 0.15 |
| | TFS | **14.11** | **18.63** | **2.53** | **0.69** | **0.28** |
| 228.77 img/s | Vanilla | 65.15 | 68.64 | 2.13 | 0.40 | 0.01 |
| | TFS | **31.24** | **36.63** | **2.26** | **0.60** | **0.10** |

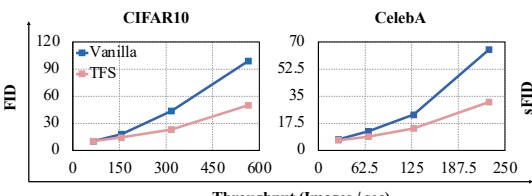
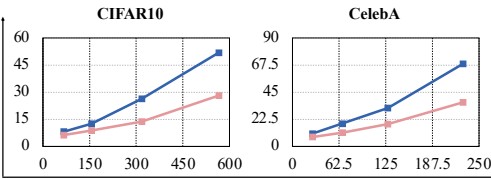

Figure 2: Comparison of throughput vs. sample quality and diversity on CIFAR-10 (left) and CelebA (right). The table (top) reports FID, sFID, IS, Precision, and Recall across throughput levels for Vanilla and TFS). The plots (bottom) visualize FID (right) and sFID (left) for DDIM, highlighting how TFS (red) compares against Vanilla DDIM (blue).

*components and hyperparameters unchanged* (Karras et al., 2022). All methods, including ours and baselines, are trained from scratch starting from the model's initialization to ensure a fair comparison. During training, we also ensure that wall-clock time matches that of the original model. Because our method yields $K \times$ throughput, it effectively provides $K \times$ more gradient updates under the same time budget. Notably, **Timestep-Fusion Scheduling (TFS)** requires no additional training. It is applied purely at inference time by combining already trained models with $K = 1, 2, 4$, enabling adaptive fusion without altering training cost. Training and optimization details are provided in the Appendix.

**Metrics** We assess perceptual quality and distributional coverage along five complementary axes. First, we report the Inception Score (IS) (Salimans et al., 2016) and Fréchet Inception Distance (FID) (Heusel et al., 2017) to jointly capture sample diversity and global visual fidelity. Second, we compute the spatial FID (sFID), which measures the Fréchet distance over intermediate feature maps that preserve spatial structure, thereby providing a more sensitive evaluation of local detail and geometric fidelity (Nash et al., 2021; Dhariwal & Nichol, 2021). Finally, we measure precision (i.e, fraction of generated samples that lie on or near the true data manifold), and recall (i.e, fraction of the real data manifold covered by generated outputs) to quantify the extent to which FuseBatch preserves individual sample fidelity while still covering the full diversity of the true data manifold. (Kynkäänniemi et al., 2019; Sajjadi et al., 2018).

### 4.1 APPLICATION TO STANDARD MODELS WITH TFS

We first apply FuseBatch to standard DDPMs and extend it to their class-conditional variants. With a fixed fusion factor (e.g., $K = 2$ or $K = 4$), throughput improves proportionally while adding only a minimal parameter overhead (at most 0.29%). Importantly, we find that when $K$ is small (e.g., $K = 2$), the quality drop

Table 2: Class-conditional CIFAR-10.

| Model | FID↓ | sFID↓ | IS↑ | Prec.↑ | Rec.↑ |
|---|---|---|---|---|---|
| Baseline(DDPM) | 4.07 | 4.37 | 9.04 | 0.66 | 0.56 |
| FuseBatch ($K=2$) | 7.56 | 5.65 | 8.65 | 0.65 | 0.54 |

is modest: FID reflects a slight degradation, while sFID, IS, Precision, and Recall remain largely unaffected, indicating preserved diversity and distributional coverage. For instance, on CelebA, FID increases from 3.51 to 4.73 but sFID remains nearly unchanged (6.27 → 6.98). Under the class-conditional DDPM on CIFAR-10, the quality degradation at $K = 2$ mirrors that of the unconditional setting, confirming that FuseBatch applies seamlessly to conditional generation with acceptable trade-offs (Table 2). The stability of IS and Precision suggests that although generated samples diverge slightly from the original distribution, perceptual quality, fidelity, and diversity are preserved.

Table 3: Performance of FB-AE imported in LDM for $K = 2, 4$. Reports parameter overhead, inference throughput, MAC reduction, and sample quality metrics (FID ↓, sFID ↓, IS ↑, Precision ↑, Recall ↑). Experiments are conducted on the CelebA-HQ dataset.

| Method | CelebA-HQ | | | | | | | |
|---|---|---|---|---|---|---|---|---|
| | Parameter | Throughput | MAC | FID↓ | sFID↓ | IS↑ | Precision↑ | Recall↑ |
| LDM(baseline) | 362.3M | 0.54 | 96.38T | 11.58 | 12.68 | 3.12 | 0.68 | 0.34 |
| FuseBatch(K=2) | +159.8K (0.04%) | 1.08 (×2.00) | 48.19T (50.0%) | 12.90 | 13.74 | 2.96 | 0.66 | 0.33 |
| FuseBatch(K=4) | +319.6K (0.08%) | 0.54 (×4.00) | 24.09T (25.0%) | 14.57 | 16.79 | 2.78 | 0.69 | 0.24 |

These results highlight FuseBatch's effectiveness as a general strategy for boosting throughput while maintaining sample quality in low-$K$ regimes.

However, as $K$ grows larger, the trade-off becomes less favorable, motivating the need for an adaptive strategy. For example, Table 1 shows that on CIFAR-10, FID rises from 3.41 to 5.93 at $K = 2$, but to 11.20 at $K = 4$. A similar trend is observed on CelebA, where throughput gains are consistent yet quality degradation becomes pronounced. To overcome this limitation, we introduce **Timestep-Fusion Scheduling (TFS)**, a purely inference-time policy that adaptively allocates $K$ across timesteps. We allocate 50% of steps to $K{=}4$, 25% to $K{=}2$, and the remaining 25% to $K{=}1$.[2] For DDPM, whose trajectories proceed coarse-to-fine (Karras et al., 2022; Yang et al., 2023), the $K{=}4/2/1$ sets are arranged from early to late so that $K$ decreases over time. This schedule achieves a favorable trade-off: throughput improves by approximately ×1.7 while FID degrades only modestly (e.g., 3.41 → 4.77). Notably, on CIFAR-10, IS improves (9.20 → 9.29) and Recall rises (0.58 → 0.60), while on CelebA, sFID is reduced (6.27 → 6.00), even surpassing the vanilla baseline. These improvements suggest that adaptive fusion not only preserves efficiency but can also enhance distributional alignment. Qualitative examples in Appendix confirm that TFS maintains perceptual fidelity and diversity while still delivering near-multiplicative acceleration.

### 4.2 EXTENSION TO STEP-REDUCED MODELS

We further extend FuseBatch to step-reduction frameworks such as DDIM and Flow Matching (FM). Since these methods shorten the sampling trajectory, they are naturally complementary to our approach, which increases the number of samples generated per step. Consistent with the standard DDPM setting, we find that small fusion factors (e.g., $K = 2$) deliver substantial acceleration with only modest quality impact. For example, on CIFAR-10 under FM, FID increases from 2.74 to 6.72 at $K = 2$, while other metrics such as IS, sFID, Precision, and Recall remain largely stable (Table 1). This indicates that FuseBatch preserves diversity and perceptual quality even when combined with step-reduction, validating its compatibility across different sampling paradigms.

However, quality degradation becomes more severe as $K$ grows larger, mirroring the trend observed in the vanilla DDPM setting. For instance, applying $K = 4$ under DDIM or FM leads to sharper rises in FID despite continued throughput gains, revealing that naïvely fixing $K$ across all timesteps is suboptimal. To address this limitation, we integrate **Timestep-Fusion Scheduling (TFS)** into step-reduced samplers to preserve fidelity while still delivering significant acceleration (Table 1). DDIM exhibits the same alignment between timesteps and importance, and therefore adopts the same scheduling strategy as DDPM. In contrast, Flow Matching lacks a sequential timestep order. Therefore, we instead rank timesteps by scheduler interval and apply the same global allocation in order of increasing interval size (smaller interval → larger $K$, larger interval → smaller $K$). To achieve approximately 2× throughput, we maintain the same ratio of steps across different $K$ values as used in DDPM. In both cases, adaptive scheduling not only restores the balance between efficiency and quality but also produces secondary gains (e.g., improved IS or sFID in certain regimes). These results demonstrate that FuseBatch, when combined with TFS, synergizes with step-reduction to unlock compounding efficiency gains while avoiding the sharp quality penalties of large fixed-$K$ fusion.

---

[2]This yields approximately 2× throughput since the effective cost is $\frac{1}{4} \times 0.5 + \frac{1}{2} \times 0.25 + 1 \times 0.25 = 0.5$.

## 4.3 Unlocking Throughput Limits

In this section, we demonstrate that **FuseBatch** surpasses the practical throughput ceilings of conventional step-reduction methods. This is possible because, as shown in Sec. 4.2, FuseBatch doubles inference throughput with minor quality drop using **Timestep-Fusion Scheduling (TFS)**, thereby enabling the model to operate at higher throughput settings than conventional approaches. To highlight this advantage, we integrated TFS into DDIM and measured image fidelity at 50, 20, and 10, 5 sampling steps. We then compared these results to those of a standard DDIM baseline operating at the same throughputs in order to assess whether step reduction alone can achieve comparable speed without severe quality degradation. As shown in Fig. 2, FuseBatch achieves superior performance to the conventional approach across nearly all metrics, maintaining higher sample quality at equal throughput settings. The baseline's FID rises sharply once pushed beyond its practical ceiling, FuseBatch exhibits a much more gradual decline. These findings confirm that FuseBatch unlocks new practical limits for high-throughput diffusion inference. Corresponding quantitative results are provided in the Appendix.

## 4.4 Extension to Latent Models with Higher Resolution

Having demonstrated that FuseBatch can be seamlessly integrated with step-reduction methods such as DDIM and Flow Matching, we next investigate its applicability to latent diffusion models (LDMs). This direction shows that our method is not limited to pixel-space denoising but extends naturally to latent-space generative frameworks, underscoring its compatibility with recent scalable diffusion models (e.g Stable Diffusion 1.5). To this end, we evaluate FuseBatch on CelebA-HQ using an LDM backbone, which operates in compressed latent space and thus represents a widely adopted high-resolution setting.

As reported in Table 3, FuseBatch preserves its throughput advantage with only modest degradation in sample quality. Notably, when scaling the fusion factor to $K = 4$, the decline in FID is far less severe on CelebA and CelebA-HQ than on CIFAR-10 (Table 1). This reflects an intrinsic resolution effect: encoding four images into a $32 \times 32 \times 3$ space imposes a heavier bottleneck than in $64 \times 64 \times 3$ or latent $256^2$ settings. Although high-resolution images contain more absolute information, their sparsity reduces per-pixel density, making larger fusion factors more viable (Olshausen & Field, 1996; Simoncelli & Olshausen, 2001; Rombach et al., 2022).

Unlike pixel-space models, **Timestep-Fusion Scheduling (TFS)** cannot be applied in this setting. Because FB-AE wraps the entire denoising process, fusion and unfusion are tied to the autoencoder interface, leaving no hook to vary $K$ across timesteps. Once a fusion factor is chosen, it must remain fixed throughout the trajectory. Nevertheless, FuseBatch still performs strongly in LDMs, showing that stable multi-sample fusion alone is sufficient for high-resolution settings. From these results, two key insights emerge: (i) FuseBatch scales effectively to latent models and thus to high-resolution generators, and (ii) larger fusion factors ($K > 2$) are increasingly viable as resolution grows. Together, they demonstrate FuseBatch's strength in keeping pace with the trend toward larger diffusion models for high-fidelity image synthesis (Peebles & Xie, 2023).

## 5 Conclusion

In this paper, we address the throughput limitation of diffusion models by shifting focus from reducing the cost of a single trajectory to generating multiple images in one forward pass. To realize this idea, we propose **FuseBatch**, introducing lightweight fusion–unfusion operators instantiated as **FB-UNet** for pixel-space models and **FB-AE** for latent-space models. We show that FuseBatch improves throughput with negligible overhead, applies effectively to DDPM, while remaining orthogonal to step-reduction methods(e.g DDIM, flow matching). To further balance speed and fidelity, we propose **Timestep-Fusion Scheduling (TFS)**, which adapts the fusion factor across timesteps, thereby enabling FuseBatch to surpass the baseline at comparable throughput settings. Extending to latent diffusion, FuseBatch proves especially effective in high-resolution settings where larger fusion factors are viable. For future work, we expect FuseBatch to provide a foundation for adaptive schedules, stronger architectures for large-$K$ fusion, and integration with future generative frameworks, paving the way toward faster and more scalable diffusion-based generation.

**Reproducibility Statement**   We have taken several steps to ensure the reproducibility of our work. In the Appendix, we provide detailed model configurations and training settings. Our experimental setup closely follows the guidelines established in prior works on diffusion acceleration, so reproducing these baselines directly translates to reproducing our method without additional difficulty. Most importantly, we will submit the full source code as supplementary material, enabling researchers to reproduce our results and extend our approach with minimal effort.

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

## A    SUPPLEMENTAL MATERIALS

### A.1    IMPLEMENTATION DETAILS

We describe here the training and inference configurations used across all datasets and model families. For CIFAR-10 DDPM, we follow the protocol of Ho et al. (Ho et al., 2020), while CelebA DDPM and the DDIM samplers adopt the settings of Song et al. (Song et al., 2020a). For Flow Matching on CIFAR-10, we replicate Lipman et al. (Lipman et al., 2024); for CelebA, no official baseline exists, so we define its configuration consistently with the rest of our experiments. Across all cases, the same learning rate schedule, noise schedule, and sampling procedure are maintained regardless of the fusion factor $K$. Evaluation is performed on 50k generated samples using the implementation of Dhariwal et al. (Dhariwal & Nichol, 2021), ensuring that any performance differences can be attributed solely to the FuseBatch modules. The detailed configuration is provided in Table 4.

Table 4: Training and inference configurations of DDPM, DDIM, and Flow Matching on CIFAR-10 and CelebA, including model architecture parameters, diffusion sampling schedules, and optimization hyperparameters.

|  | CIFAR-10 | | | CelebA | | |
|---|---|---|---|---|---|---|
|  | DDPM | DDIM | Flow Matching | DDPM | DDIM | Flow Matching |
| Channels | 128 | 128 | 128 | 128 | 128 | 128 |
| Channel Multiplier | 1,2,2,2 | 1,2,2,2 | 2,2,2 | 1,1,2,2,4,4 | 1,1,2,2,4,4 | 2,2,4,4 |
| Number of Heads | 1 | 1 | 1 | 16,16,32,32,64,64 | 16,16,32,32,64,64 | 1 |
| Activation Function | SiLU | SiLU | SiLU | SiLU | SiLU | SiLU |
| Diffusion Steps | 1000 | 50 | 50 | 1000 | 50 | 50 |
| Noise Schedule | linear | linear | – | linear | linear | – |
| Beta Start | 0.0001 | 0.0001 | – | 0.0001 | 0.0001 | – |
| Beta End | 0.02 | 0.02 | – | 0.02 | 0.02 | – |
| Dropout | 0.1 | 0.1 | 0.3 | 0.0 | 0.0 | 0.3 |
| Learning Rate | 2e-4 | 2e-4 | 1e-4 | 2e-4 | 2e-4 | 1e-4 |
| Batch Size | 128 | 128 | 64 | 32 | 32 | 64 |

### A.2    ANALYSIS OF THROUGHPUT-CENTRIC DESIGN

**Comparison with Naive $K = 4$ Baseline.**    To evaluate the feasibility of FuseBatch under higher fusion factors, we compare our method at $K = 4$ against a straightforward patch-based baseline. Using the same model architectures and dataset configurations described in Section A.1, we design a naive multi-sample strategy where each noisy input is first downscaled to one-quarter resolution, concatenated into a $2 \times 2$ grid, processed once by the diffusion model, and then split and upscaled back to full resolution (Li et al., 2024). This simple approach, however, severely degrades generation quality. As illustrated in Table 5, the patch-based method introduces artifacts, unnatural boundaries, and blurred details, particularly on facial features and textured backgrounds, while FuseBatch preserves fine textures and color consistency.

Quantitative results in Table 6 further confirm this gap. When applied to a standard DDPM, the patch-based method suffers a dramatic quality drop—on CIFAR-10, FID deteriorates from 18.27 (FuseBatch) to 91.71, and IS falls from 7.41 to 4.55. On CelebA, FuseBatch achieves an FID of 9.50 versus 26.07 for the grid approach, and an IS of 2.58 compared to 2.37. These findings demonstrate that our fusion–unfusion modules not only preserve and disentangle multi-sample information more effectively than naive patch compositing but also scale reliably to higher throughput settings. Together, the qualitative and quantitative evidence underscores the scalability of FuseBatch and its potential to unlock further efficiency gains without compromising visual fidelity.

**Comparison to Throughput-Matched Small Models.**    Beyond comparisons to naive compositing, we also evaluate whether FuseBatch simply mimics the effect of reducing model size to match a desired throughput. To this end, we construct two reduced-FLOPs U-Net variants trained on CIFAR-10 at $K = 2$. The first is a *channel-reduced* model, which decreases the number of feature maps per layer by modifying the channel multipliers. The second is a *depth-reduced* model, which shortens the network depth by removing residual blocks. These models are designed to operate under comparable computational budgets as FuseBatch, thereby serving as throughput-matched baselines.

Table 5: Qualitative comparison of FuseBatch ($K = 4$) versus a naive $2 \times 2$ patch-based baseline on CIFAR-10 and CelebA. FuseBatch maintains sharper details and avoids mosaic artifacts.

| CIFAR-10 | | CelebA | |
|---|---|---|---|
| FuseBatch (K=4) | Patch-Based | FuseBatch (K=4) | Patch-Based |

Table 6: Quantitative performance of FuseBatch ($K = 4$) compared with a naive patch-based baseline on CIFAR-10 and CelebA. Metrics include FID, sFID, IS, Precision, and Recall. FuseBatch significantly outperforms the naive approach.

| | CIFAR-10 | | | | | CelebA | | | | |
|---|---|---|---|---|---|---|---|---|---|---|
| Method | FID↓ | sFID↓ | IS↑ | Precision↑ | Recall↑ | FID↓ | sFID↓ | IS↑ | Precision↑ | Recall↑ |
| FuseBatch (K=4) | 18.27 | 6.54 | 7.41 | 0.56 | 0.55 | 9.50 | 10.66 | 2.58 | 0.70 | 0.38 |
| Patch-Based | 91.71 | 52.16 | 4.55 | 0.35 | 0.07 | 26.07 | 37.34 | 2.37 | 0.36 | 0.12 |

Detailed configurations are listed in Table 7, and quantitative results are reported in Table 8. While both reduced models achieve fewer FLOPs and lower throughput than the baseline DDPM, they suffer clear degradation in generative quality. On CIFAR-10, FID rises from 5.93 (baseline) to 7.59 (channel-reduced) and 6.55 (depth-reduced), with corresponding declines in IS and recall. The channel-reduced model shows the most pronounced drop, reflecting the loss of representational capacity when latent dimensionality is heavily compressed. The depth-reduced model fares slightly better but still lags behind FuseBatch.

These results highlight a key distinction: reducing network capacity to save FLOPs inevitably shrinks the latent space available for each sample, limiting the model's ability to store and recover fine-grained image information. By contrast, FuseBatch maintains the full backbone capacity while distributing multiple inputs through shared computation, thereby preserving fidelity and diversity even under higher throughput demands. Although fusion introduces a mild bottleneck at the encoding stage, it nonetheless outperforms simple FLOPs-matched capacity-reduced variants. In other words, FuseBatch achieves efficiency by increasing per-pass yield without sacrificing the richness of the latent representation, whereas smaller models pay a steep price in generative quality for reduced cost.

**Extension to Latent Diffusion Models.** Finally, we extend FuseBatch to the Latent Diffusion Model (LDM) framework trained on CelebA-HQ in order to test whether our approach generalizes to high-resolution generative settings. Unlike pixel-space diffusion models, LDMs operate in a compressed latent space, which makes them a widely adopted backbone for large-scale and high-resolution synthesis. Within this setup, we consider two alternative integration points for our fusion modules. The first design, denoted *FB-UNet*, inserts fusion directly inside the denoising UNet backbone. In this case, multiple noisy latent samples are combined and processed jointly during the denoising steps. However, because fusion occurs inside the denoising network, small latent perturbations introduced during compression are often amplified when decoded back to pixel space, which can lead to visible artifacts and degraded fidelity. The second design, denoted *FB-AE*, instead applies fusion at the autoencoder stage. Here, noisy inputs are fused before decoding, and unfusion occurs after the autoencoder reconstruction. This approach avoids the amplification issue seen in FB-UNet, since fusion is performed closer to pixel space where minor perturbations are less destructive. As summarized in Table 9, the difference between the two strategies is substantial. FB-AE achieves both higher throughput (almost doubling inference speed) and stronger fidelity across quality metrics,

Table 7: Configuration details for CIFAR-10 DDPM baseline, channel-reduced, and depth-reduced variants.

| | CIFAR-10 (DDPM) | | |
|---|---|---|---|
| | Baseline | Channel-Reduced | Depth-Reduced |
| Channels | 128 | 128 | 128 |
| Channel Multiplier | 1,2,2,2 | 1,1,1,2 | 1,1,2 |
| Number of Heads | 1 | 1 | 1 |
| Activation Function | SiLU | SiLU | SiLU |
| Diffusion Steps | 1000 | 1000 | 1000 |
| Noise Schedule | linear | linear | linear |
| Beta Start | 0.0001 | 0.0001 | 0.0001 |
| Beta End | 0.02 | 0.02 | 0.02 |
| Dropout | 0.1 | 0.1 | 0.1 |
| Learning Rate | 2e-4 | 2e-4 | 2e-4 |
| Batch Size | 128 | 128 | 128 |

Table 8: Performance of CIFAR-10 DDPM baseline, channel-reduced, and depth-reduced models. FuseBatch achieves higher quality at similar throughput compared to reduced models.

| Method | CIFAR-10 (DDPM) | | | | | | |
|---|---|---|---|---|---|---|---|
| | Throughput | MAC | FID↓ | sFID↓ | IS↑ | Precision↑ | Recall↑ |
| Baseline | 3.38 | 3.07T | 5.93 | 4.58 | 8.67 | 0.63 | 0.58 |
| Channel-Reduced | 2.75 | 3.26T | 7.59 | 4.78 | 8.25 | 0.62 | 0.55 |
| Depth-Reduced | 2.54 | 3.72T | 6.55 | 4.65 | 8.47 | 0.63 | 0.57 |

whereas FB-UNet shows clear performance degradation. These results demonstrate that FuseBatch can be effectively adapted to high-resolution latent diffusion pipelines, and that careful placement of the fusion–unfusion modules is crucial to preserve fidelity while reaping efficiency gains. Qualitative results of FB-AE with various K and FB-UNet implemented into LDM is shown in Figure 11.

## A.3 ABLATION STUDY

We conduct an ablation study to investigate the effect of kernel size in the fusion stage. As shown in Table 10, increasing the kernel size leads to degraded quality across FID, sFID, and IS. This effect arises because larger kernels introduce stronger entanglement of spatial information across inputs, which in turn hampers effective disentanglement during unfusion. These results indicate that smaller kernels are preferable for stable fusion and higher fidelity.

## A.4 QUALITATIVE RESULTS

We present qualitative evidence to complement the quantitative metrics. First, we provide random samples generated from DDPM, DDIM, and Flow Matching models under different fusion factors $K = 1, 2, 4$ along with TFS. As shown in Table 12, 13, FuseBatch preserves structural coherence and perceptual fidelity across datasets and samplers, highlighting the robustness of our framework across diverse generative settings.

We further examine how image fidelity scales with throughput under different DDIM sampling steps. Table 14, 15 shows samples from TFS on CIFAR-10 and CelebA at timesteps 5, 10, 20, 50. We also include qualitative results of FuseBatch ($K = 2$). Despite aggressive reduction of sampling steps, the generated images remain competitive in quality, demonstrating graceful scaling toward real-time inference.

Table 9: FuseBatch in Latent Diffusion Models on CelebA-HQ. FB-AE achieves both higher throughput and better quality than FB-UNet.

| Method | | Parameter | Throughput | MAC | FID↓ | sFID↓ | IS↑ | Precision↑ | Recall↑ |
|---|---|---|---|---|---|---|---|---|---|
| | | **CelebA-HQ** | | | | | | | |
| LDM(baseline) | | 362.3M | 0.54 | 96.38T | 11.58 | 12.68 | 3.12 | 0.68 | 0.34 |
| FB-UNet(K=2) | | +159.8K (0.04%) | 1.07 (×1.98) | 48.67T (50.5%) | 17.62 | 17.97 | 3.00 | 0.60 | 0.30 |
| FB-AE(K=2) | | +122.0K (0.03%) | 1.08 (×2.00) | 48.19T (50.0%) | 12.90 | 13.74 | 2.96 | 0.66 | 0.33 |

Table 10: Ablation on fusion kernel size. Larger kernels increase spatial entanglement, degrading sample quality. Smaller kernels (e.g., size 3) yield the best performance across FID, sFID, and IS.

| Kernel | | FID↓ | sFID↓ | IS↑ | Precision↑ | Recall↑ |
|---|---|---|---|---|---|---|
| 3 | | 5.93 | 4.58 | 8.67 | 0.63 | 0.58 |
| 5 | | 6.94 | 4.74 | 8.58 | 0.61 | 0.59 |
| 7 | | 7.47 | 4.74 | 8.33 | 0.62 | 0.58 |

## USE OF LARGE LANGUAGE MODELS

In this paper, large language models (LLMs) were used solely for refining the writing style and checking grammar. No aspect of the conceptual development, experimental design, implementation, or analysis relied on the use of an LLM. All scientific contributions are entirely the work of the authors.

Table 11: Qualitative comparison of LDM (vanilla), FB-UNet, and FB-AE with different fusion factors.

| LDM (vanilla) | FB-UNet (K=2) | FB-AE (K=2) | FB-AE (K=4) |
|---|---|---|---|

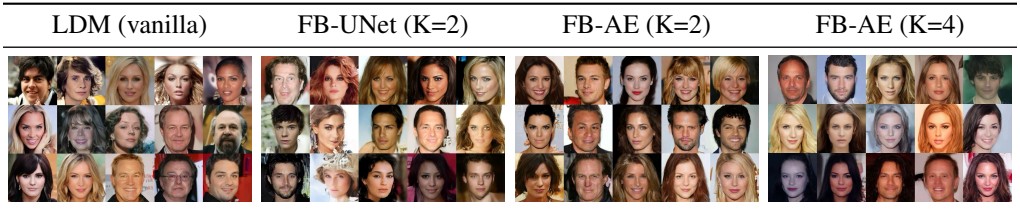

Table 12: Randomly generated samples from DDPM, DDIM (T=50), and Flow Matching on **CIFAR-10** for $K = 1, 2, 4$ and TFS. FuseBatch maintains coherence and diversity across different samplers.

| K | DDPM | DDIM (T=50) | FM |
|---|---|---|---|

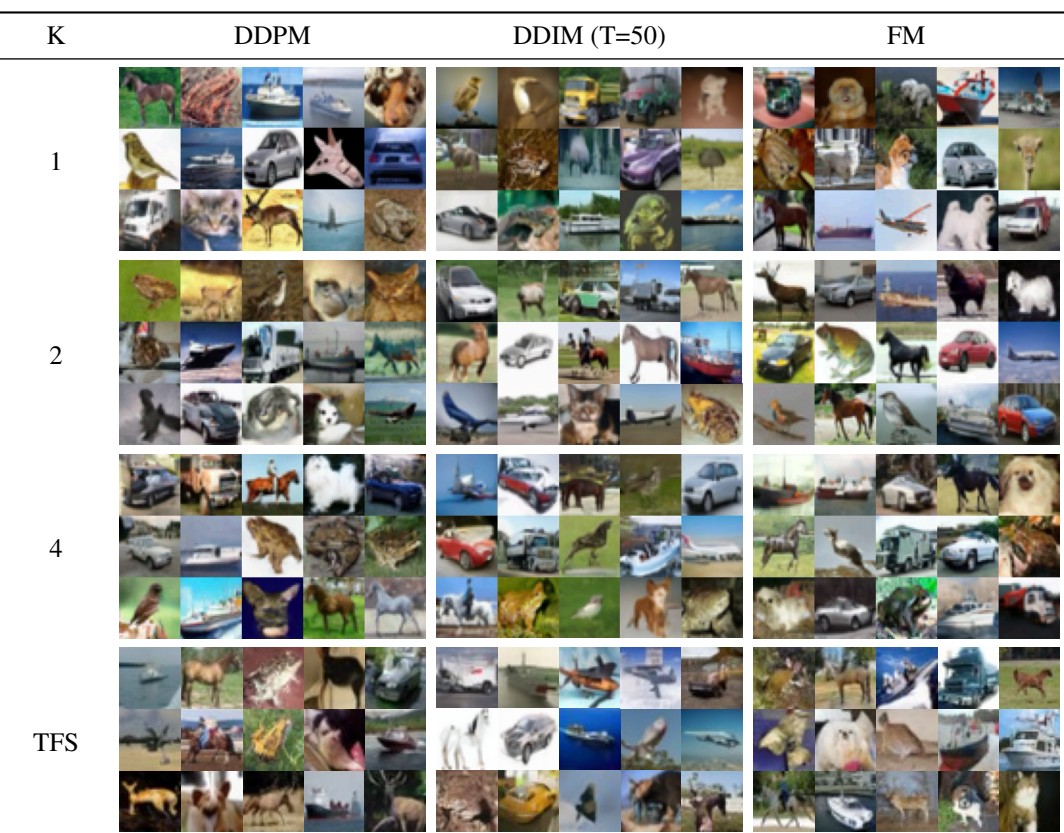

Table 13: Randomly generated samples from DDPM, DDIM (T=50), and Flow Matching on **CelebA** for $K = 1, 2, 4$ and TFS. Results confirm that FuseBatch preserves sample fidelity and diversity across datasets.

| K | DDPM | DDIM (T=50) | FM |
|---|---|---|---|

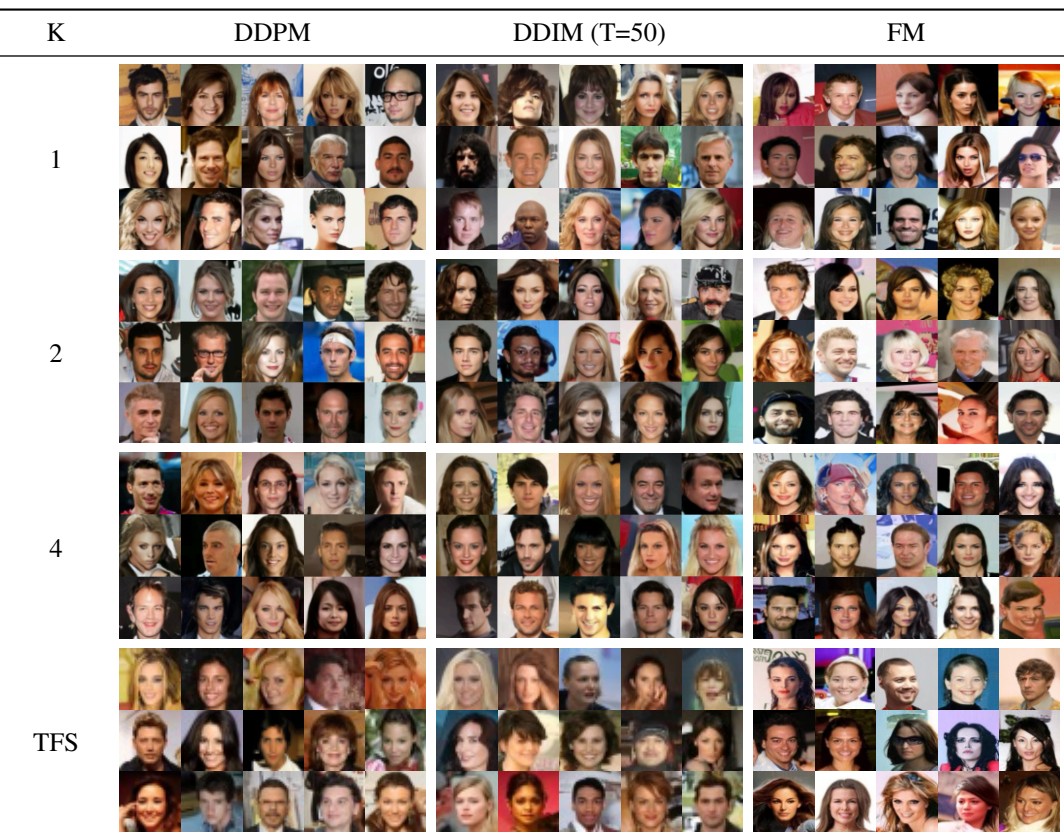

Table 14: Qualitative scaling with throughput on **CIFAR-10**: random samples from FuseBatch (K=2) and TFS using DDIM with timesteps 10, 20, 50, and 100. Throughput ranges from 66.80–565.03 img/s.

| Throughput | 565.03 img/s | 317.89 img/s | 156.56 img/s | 66.80 img/s |
|---|---|---|---|---|

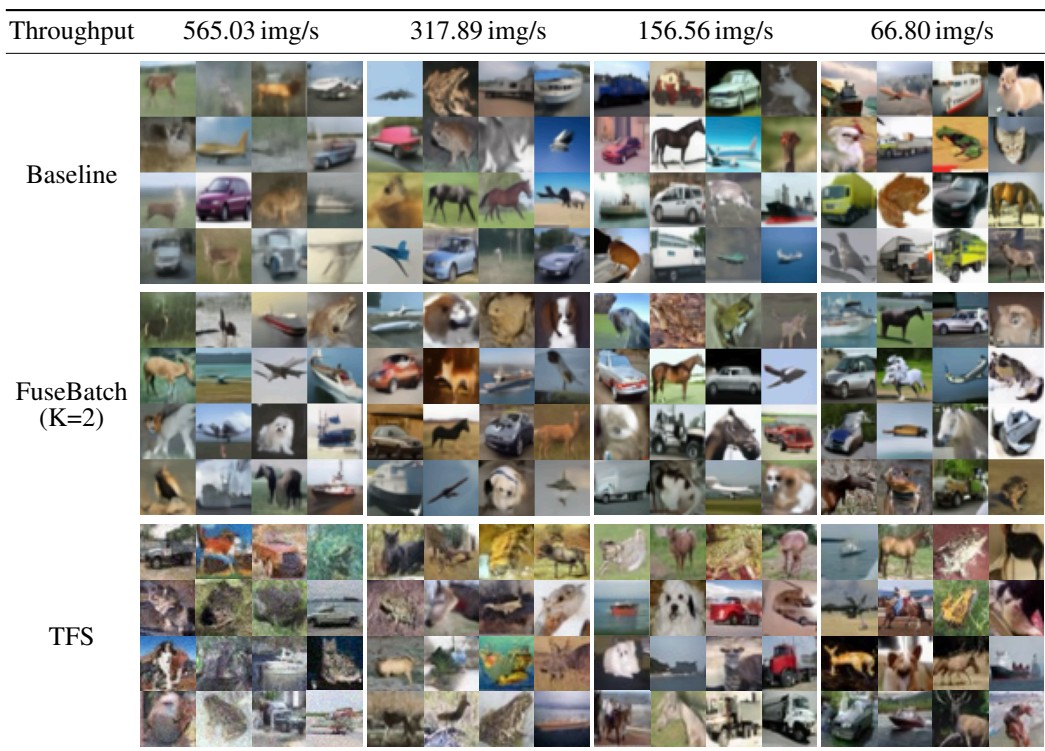

Baseline

FuseBatch (K=2)

TFS

Table 15: Qualitative scaling with throughput on **CelebA**: random samples from FuseBatch (K=2) and TFS using DDIM with timesteps 10, 20, 50, and 100. Throughput ranges from 26.83–228.77 img/s.

| Throughput | 228.77 img/s | 127.92 img/s | 66.90 img/s | 26.83 img/s |
|---|---|---|---|---|

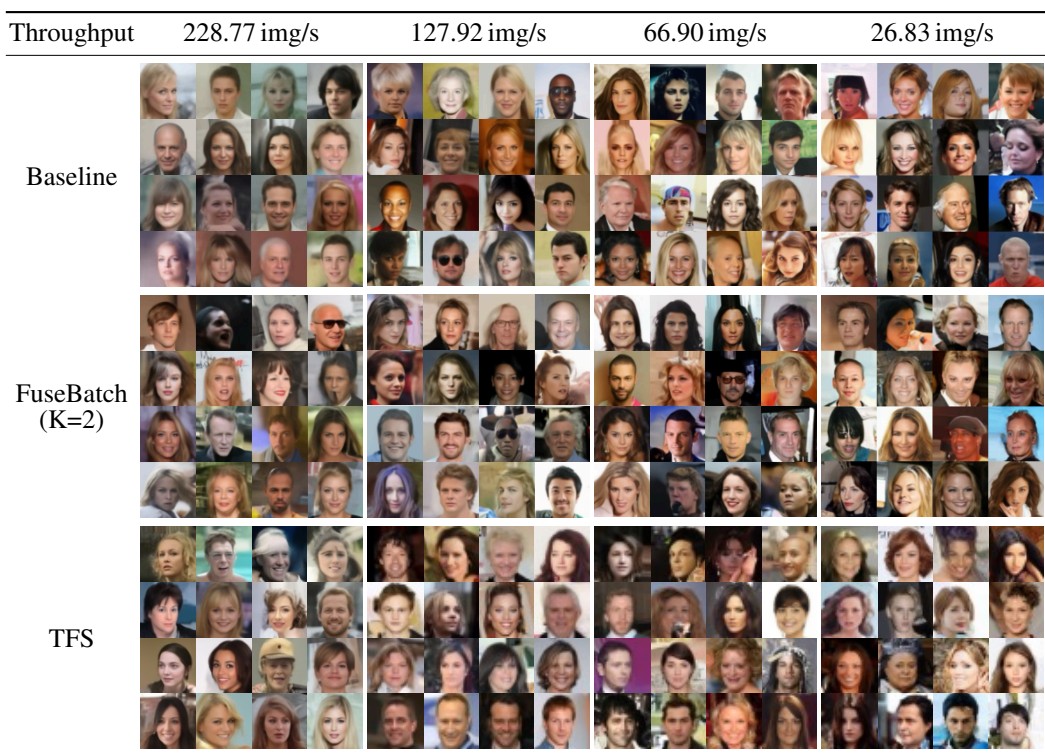

Baseline

FuseBatch (K=2)

TFS

