# OpenReview forum: "FuseBatch: Unlocking the Potential of Diffusion Models in Throughput Perspective"
_ICLR.cc/2026/Conference — Submitted to ICLR 2026_

### Official Review · Reviewer_FNDa · 2025-10-30

**Soundness:** 2
**Presentation:** 3
**Contribution:** 2
**Rating:** 2
**Confidence:** 5

**Summary:**

The slow sampling speed of diffusion models is due to progressive denoising, requiring each image to traverse an entire time step. Mainstream acceleration methods focus on reducing the number of steps, but further reduction after a dozen or so steps may significantly degrade image quality, reaching a plateau in marginal gains. This paper proposes a different approach called  FuseBatch. Instead of reducing the time per image, it aims to generate multiple images in a single forward pass, maximizing throughput.

**Strengths:**

1. The proposed fusebatch idea is interesting. Previous diffusion methods have focused on reducing the sampling step or the time required for each step. This paper proposes a new approach: accelerating the throughput of the entire batch by fusing and then decomposing multiple images during sampling.
2. The proposed method is compatible with DDIM and Flow Matching.
3. The paper is written clearly.

**Weaknesses:**

1. The paper lacks several essential experiments. The authors only conducted experiments on small-scale class-conditional image generation datasets such as CIFAR-10 and CelebA. As a training-free acceleration method, it is necessary to demonstrate its practical acceleration performance on mainstream text-to-image generation models (such as SDXL).

2. The experiments in the paper lack necessary baseline comparisons. I doubt whether the proposed method can achieve better results than simple training-free approaches such as DeepCache.

3. Can the proposed method be compatible with DPM-Solver?

4. Both the method description and experimental sections of the paper are based on U-Net-based diffusion models. Considering that the current state-of-the-art diffusion models mostly adopt the DiT architecture, it is necessary to demonstrate whether the proposed method can also accelerate DiT-based diffusion models.

**Questions:**

Please see the weakness.

---

> ### Author Response · Authors · 2025-11-22
>
> **Positioning of Our Contribution**
>
> We would like to clarify the intended positioning of our work. The purpose of this paper is to introduce a complementary acceleration direction based on generating multiple samples within a single forward pass, rather than to compete for state of the art performance. Although achieving state of the art performance is not the objective of this paper, we emphasize that our approach is not limited to a narrow class of models. We applied our method to DDPMs, DDIM, Flow Matching, and latent diffusion models, and during the rebuttal period we further confirmed that the same mechanism can be applied to DiT-style transformer backbones as well. These results show that our method is compatible with the major diffusion frameworks that underlie many current state of the art systems, illustrating its potential to extend naturally to stronger and larger architectures. We also note that recent diffusion studies accepted to top venues([1],[2],[3]), validate their key mechanisms primarily on CIFAR-10 or CelebA HQ scale models. This reflects a well established practice for proof of concept evaluation in diffusion research and indicates that demonstrating a new methodological direction on these standard backbones does not diminish the significance of the contribution.
>
> [1] Zheng, Xingyu, et al. "Binarydm: Accurate weight binarization for efficient diffusion models.” ICLR 2025.
>
> [2] Mark, Wang Zhe, et al. "Distribution-Aware Diffusion Model Quantization via Distortion Minimization.” ICLR 2025.
>
> [3] Karczewski, Rafał, et al. "Diffusion Models as Cartoonists: The Curious Case of High Density Regions.” ICLR 2025.

---

> ### Author Response · Authors · 2025-11-22
>
> **[W1] High-Resolution and SDXL Scalability**
>
> Regarding your concern about the absence of results on large-scale text-to-image models such as SDXL, we apologize for any confusion. As noted in Section 4 (Implementation Details), our framework requires **model pretraining** because the fusion and unfusion modules must be co-trained with the backbone. Therefore, our method is **not a training free acceleration technique**, and training a model of SDXL scale from scratch is infeasible under our resource constraints. For this reason, we were unable to conduct full experiments on SDXL level models within the scope of this submission.
>
> Instead, we aimed to provide empirical and theoretical evidence demonstrating that our mechanism becomes more reliable as the dimensionality of the data increases. Specifically, higher resolution image spaces exhibit significantly stronger sparsity and manifold structure, which increases the tolerance to larger fusion factors K. We further clarify the intuition behind this phenomenon below.
> Consider first a simple two dimensional multi modal distribution before extending the argument to high dimensional image manifolds. Suppose the data contain two modes, one centered at (−1, −1) and the other at (+1, +1). If two samples are drawn and added elementwise, the result is either (−2, −2), (0, 0), or (2, 2). These three outcomes remain well separated, so the fused representation does not collapse information but instead preserves a structured relationship between the inputs. This illustrates why fusion can be effective even when performed with a simple additive operation. This example also provides intuition for why unfusion is possible. A fused value such as (0, 0) could theoretically arise from infinitely many different pairs in the continuous plane. However, we do not seek a decomposition in the full continuous space. We only consider decompositions that lie on the real data distribution. This constraint eliminates the vast majority of mathematically possible pairs, leaving only a very small number of valid combinations and often a single one that is consistent with the manifold. This intuition extends to natural images as they are known to lie on extremely sparse low dimensional manifolds([4],[5],[6]) while exhibiting highly multimodal structure([7],[8]), which makes the fused representation retain separable information and keeps both fusion and unfusion inherently tractable. This property becomes even stronger in high dimensional or high resolution image spaces, where the manifold structure is more pronounced([6],[9]) and invalid decompositions fall off the manifold immediately. As a result, the fused representation almost uniquely determines the original inputs, and the unfusion module only needs to resolve a small and highly constrained ambiguity rather than searching over arbitrary combinations.
>
> We therefore believe that, although full SDXL scale experiments are beyond our computational budget, the underlying principles of our method naturally extend to high resolution and large scale text to image systems.
>
> [4]Zhu, Jun-Yan, et al. "Generative visual manipulation on the natural image manifold.” ECCV 2016.
>
> [5]Rifai, Salah, et al. "A generative process for sampling contractive auto-encoders.” ICML 2012.
>
> [6]Pope, Phillip, et al. "The intrinsic dimension of images and its impact on learning." ICLR 2021.
>
> [7]Zhu, Jun-Yan, et al. "Toward multimodal image-to-image translation.” NeurIPS 2017.
>
> [8]Huang, Xun, et al. "Multimodal unsupervised image-to-image translation.” ECCV 2018.
>
> [9]Zheng, Yijia, et al. "Learning manifold dimensions with conditional variational autoencoders." NeurIPS 2022.

---

> ### Author Response · Authors · 2025-11-22
>
> **[W2] Relationship to Training-Free Baselines**
>
> As discussed above, our framework is not training free because the fusion and unfusion modules require joint pretraining with the backbone. For this reason, we believe that a direct comparison with DeepCache, which is specifically designed as a training free acceleration method, is not an appropriate baseline in the context of our approach. However, we agree with the reviewer that demonstrating compatibility with existing training free methods would further strengthen the general applicability of our framework. Motivated by this insight, we investigated whether DeepCache can be applied orthogonally on top of our method.
>
> We found that the two approaches indeed operate on fundamentally different components of the diffusion process and therefore do not conflict with each other. To verify this, we applied the basic DeepCache mechanism to our fused inference pipeline and observed that it integrates without architectural modification and produces consistent acceleration gains. The experimental results are reported below.
>
> \begin{array}{|l|l|ccccccc|}
> \hline
> &&&& CIFAR10 &&&& \\\\
> \hline
> Model & Variant & MAC(G) & Throughput & FID & sFID & IS & Prec & Rec \\\\
> \hline
> Baseline & Vanilla & 606.88 & 17.70 & 6.30 & 5.65 & 8.88 & 0.65 & 0.56 \\\\
> \hline
> Baseline & +DeepCache & 390.18(64.3\\%) & 28.05(\\times1.59) & 6.90 & 6.21 & 8.89 & 0.64 & 0.56 \\\\
> \hline
> K=2 & Vanilla & 306.88(50.6\\%) & 33.48(\\times1.89) & 11.21 & 6.12 & 8.30 & 0.59 & 0.57 \\\\
> \hline
> K=2 & +DeepCache & 198.48(32.7\\%) & 51.19(\\times2.89) & 11.24 & 6.48 & 8.29 & 0.59 & 0.56 \\\\
> \hline
> K=4 & Vanilla & 154.63(25.5\\%) & 56.80(\\times3.21) & 23.72 & 7.22 & 7.31 & 0.54 & 0.55 \\\\
> \hline
> K=4 & +DeepCache & 100.43(16.6\\%) & 91.88(\\times5.19) & 23.87 & 8.11 & 7.37 & 0.52 & 0.54 \\\\
> \hline
> &&&& CelebA &&&& \\\\
> \hline
> Setting & Variant & MAC(G) & Throughput & FID & sFID & IS & Prec & Rec \\\\
> \hline
> Baseline & Vanilla & 1556.01 & 7.20 & 4.17 & 6.70 & 2.82 & 0.74 & 0.50 \\\\
> \hline
> Baseline & +DeepCache & 1075.81(69.1\\%) & 9.81(\\times1.36) & 6.07 & 9.10 & 2.79 & 0.72 & 0.45 \\\\
> \hline
> K=2 & Vanilla & 791.76(50.9\\%) & 13.25(\\times1.84) & 6.17 & 7.71 & 2.76 & 0.71 & 0.45 \\\\
> \hline
> K=2 & +DeepCache & 551.66(35.5\\%) & 16.96(\\times2.36) & 8.43 & 10.50 & 2.71 & 0.69 & 0.40 \\\\
> \hline
> K=4 & Vanilla & 400.63(25.8\\%) & 23.81(\\times3.31) & 12.07 & 11.93 & 2.69 & 0.65 & 0.38 \\\\
> \hline
> K=4 & +DeepCache & 280.63(18.0\\%) & 30.41(\\times4.23) & 14.29 & 14.03 & 2.62 & 0.63 & 0.32 \\\\
> \hline
> \end{array}
>
> These results indicate that our method is not only competitive as a standalone acceleration direction but also compatible with existing training free approaches. This suggests that the fusion based direction we propose can be combined with a broader range of diffusion acceleration techniques, further highlighting the flexibility and utility of our framework.
>
> **[W3] Compatibility with DPM-Solver**
>
> We also appreciate the reviewer’s suggestion regarding compatibility with DPM Solver. As with the previous point, we believe that demonstrating applicability to a wider class of diffusion acceleration techniques further strengthens the contribution of our work. To do so, we applied our fusion and unfusion mechanism within a DPM Solver based sampling pipeline. The integration required no architectural modification, and the fused sampling procedure remained stable across different solver settings. The empirical results are reported below.
>
> \begin{array}{|l|ccccccc|}
> \hline
> &&& CelebA &&&& \\\\
> \hline
> Model & MAC(G) & Throughput & FID & sFID & IS & Prec & Rec \\\\
> \hline
> Base & 778.00 & 14.59 &  & 7.41 & 2.90 & 0.69426 & 0.45068 \\\\
> \hline
> K=2 & 395.88(50.9\\%) & 25.86(\\times1.77) & 9.40 & 9.32 & 2.82 & 0.67728 & 0.40816 \\\\
> \hline
> K=4 & 200.32(25.7\\%) & 45.85(\\times3.14) & 15.23 & 13.46 & 2.75 & 0.61498 & 0.34856 \\\\
> \hline
> TFS & 393.63(50.6\\%) & 26.23(\\times1.80) & 8.86 & 8.50 & 2.90 & 0.65596 & 0.46344 \\\\
> \hline
> &&& CIFAR10 &&&& \\\\
> \hline
> Model & MAC(G) & Throughput & FID & sFID & IS & Prec & Rec \\\\
> \hline
> Base & 303.44 & 34.93 & 8.51 & 6.88 & 8.74 & 0.63412 & 0.55064 \\\\
> \hline
> K=2 & 153.44(50.6\\%) & 64.83(\\times1.86) & 13.93 & 7.35 & 8.19 & 0.58068 & 0.55498 \\\\
> \hline
> K=4 & 77.31(25.5\\%) & 116.64(\\times3.34) & 27.05 & 8.23 & 7.20 & 0.52276 & 0.53558 \\\\
> \hline
> TFS & 152.87(50.4\\%) & 64.94(\\times1.86) & 10.83 & 6.54 & 8.62 & 0.59064 & 0.56958 \\\\
> \hline
> \end{array}
>
> These results further indicate that our method can coexist with major acceleration techniques that target different aspects of the diffusion process. This reinforces the view that our framework introduces a complementary direction rather than replacing or competing with existing solvers.

---

> ### Author Response · Authors · 2025-11-22
>
> **[W4] Applicability to DiT Architectures**
>
> We agree with the reviewer that demonstrating compatibility with DiT based architectures, which form the backbone of many current state of the art diffusion models, would substantially strengthen the practical relevance of our method. To verify this, we conducted additional experiments applying our framework to DiT trained on CIFAR 10, using the standard conditional generation settings commonly adopted in recent works. As described below, our approach naturally supports two conditioning scenarios in the DiT architecture.
>
> First, the primary setting studied in our paper assumes that the K fused samples share the same condition. We confirm that this approach applies cleanly to DiT based diffusion architectures without any architectural changes. This setting is highly practical because it aligns with commercial generation workflows such as Midjourney, DALL E 3, Firefly, and Leonardo AI, where a single prompt is used to generate multiple candidate images. Since all fused inputs correspond to the same condition, FuseBatch simply applies the original conditional embedding to all K samples during both fusion and unfusion, requiring no modification to the conditioning pathway.
>
> Second, we also demonstrate that FuseBatch can support the more challenging case in which each of the K fused samples uses a different condition. To enable this scenario, we extend the DiT experiments by introducing a K way condition fusion layer. Each condition is encoded with its own embedding, modulated through a learned K indexed conditioning transformation, and then averaged to form a fused conditioning signal. Because class embeddings reside in a sufficiently high dimensional space, they can encode the necessary index information for each fused component, allowing the unfusion module to recover the individual conditional paths.
>
> \begin{array}{|l|l|ccccccc|}
> \hline
> Model & Variant & MAC(G) & Throughput & FID & sFID & IS & Prec & Rec \\\\
> \hline
> Baseline & - & 108.47 & 34.21 & 10.58 & 7.37 & 10.17 & 0.75 & 0.40 \\\\
> \hline
> K=2 & uni\\text{-}class & 56.41(52.0\\%) & 64.87(\\times1.90) & 14.89 & 8.49 & 8.87 & 0.69 & 0.37 \\\\
> \hline
> K=2 & multi\\text{-}class & 56.41(52.0\\%) & 64.61(\\times1.89) & 11.13 & 6.45 & 9.50 & 0.72 & 0.40 \\\\
> \hline
> K=4 & uni\\text{-}class & 29.56(27.3\\%) & 122.69(\\times3.58) & 23.54 & 9.65 & 8.39 & 0.61 & 0.33 \\\\
> \hline
> K=4 & multi\\text{-}class & 29.56(27.3\\%) & 122.24(\\times3.57) & 18.59 & 8.68 & 8.74 & 0.68 & 0.34 \\\\
> \hline
> TFS & - & 58.10(53.6\\%) & 122.24(\\times3.57) & 10.97 & 7.71 & 9.60 & 0.73 & 0.37 \\\\
> \hline
> \end{array}
>
> Empirically, this multi condition fusion performs even better than the single condition setting. This observation is fully consistent with our theoretical intuition. Mixing conditions from different classes corresponds to combining signals from distinct modes, which remain more separable in the fused representation and therefore simplify unfusion. In contrast, fusing samples from the same class is inherently more challenging because their representations lie in a narrower region of the manifold, creating slightly higher ambiguity. The experimental results match this reasoning closely and further validate the behavior predicted by our theoretical explanation.

---

### Official Review · Reviewer_wiEx · 2025-10-31

**Soundness:** 1
**Presentation:** 3
**Contribution:** 3
**Rating:** 2
**Confidence:** 5

**Summary:**

The paper proposes fusing images in a batch to allow a diffusion model to denoise for multiple samples simultaneously. This works very poorly in practice, but performance can be dramatically improved simply by varying the number of images that are fused depending on the progress in the denoising scheduler (less fusion for more important steps, more fusion for less important steps). This method is applied for multiple schedulers for CIFAR and CelebA, primarily in unconditional generation, using fusion-specialized UNet and autoencoder architectures.

**Strengths:**

S1. This is a novel perspective to approach something that seems intuitively desirable- reduce the tokens/latents processed per image. The idea of fusing and unfusing images is straightforward and elegant.

S2. The idea of TFS is also intuitive and elegant, simply changing the amount of fusion for different parts of the schedule.

S3. The paper clearly communicates the ideas.

**Weaknesses:**

W1. The idea is essentially a way to reduce the number of tokens/latents to process per image. While this is done by merging multiple images to a single image, it probably could be compared to simply changing the size of a patch (for DiT) or the downsampling ratio (for UNet). This seems to be the most important ablation missing, especially since we already know that we can simply increase patch size for DiT with some degradation in the quality (for example, from 19.47 to 43.01 FID, difference between DiT-XL/2 and DiT-XL/4 in Table 4 of DiT). This is reminiscent of this paper's Table 2 where for conditional generation, a fuse-batch of 2 (which speeds up computation by 2x, compared to 4x for the DiT) also doubles FID, from 4.07 to 7.56. The paper needs more convincing ablations that this method still works in the class-conditional setting, and either outperforms or is complimentary to simply changing patch size/downsample ratio.

W2. Building off of W1, the class-conditional results seem very unconvincing. My concern would be that the more you attempt to constrain the outputs, the more of a problem it would be that you are combining the denoising. If the classes or text prompts are very dissimilar, it seems this could be especially problematic. The possibility of class/prompt confusion needs to be thoroughly analyzed, and a single table with no TFS result is inadequate.

W3. The experiments are conducted on small-scale datasets in mostly unconditional settings, making it unclear if the method would work in real-world settings with higher resolution images and more conditions/constraints. This weakness is not disqualifying, but it is difficult to evaluate without even ImageNet results, let alone text to image. However, it seems natural that this method will struggle as conditions and resolution increase, so proof is needed that it does not.

W4. It is unclear if this can work in tandem with caching and quantization approaches that can achieve similar speedups, often training free, with 0 loss in performance (as measured by metrics).

**Questions:**

This is a very interesting idea, and pursues a very compelling avenue for accelerating the diffusion. However, I'm not convinced that it's better than alternatives.

1. How does this compare various approaches for pooling more aggressively on the spatial dimension?

2. How well does TFS do for class-conditional generation? What sorts of artifacts appear here, considering that K=2 has a much bigger gap than in the unconditional setting?

3. Can this work orthogonally with caching, where SOTA training-free caching methods like TaylorSeer can already support 4x+ acceleration without sacrificing quality?

---

> ### Author Response · Authors · 2025-11-22
>
> **Positioning of Our Contribution**
>
> We would like to clarify the intended positioning of our work. The purpose of this paper is to introduce a complementary acceleration direction based on generating multiple samples within a single forward pass, rather than to compete for state of the art performance. Although achieving state of the art performance is not the objective of this paper, we emphasize that our approach is not limited to a narrow class of models. We applied our method to DDPMs, DDIM, Flow Matching, and latent diffusion models, and during the rebuttal period we further confirmed that the same mechanism can be applied to DiT-style transformer backbones as well. These results show that our method is compatible with the major diffusion frameworks that underlie many current state of the art systems, illustrating its potential to extend naturally to stronger and larger architectures. We also note that recent diffusion studies accepted to top venues([1],[2],[3]), validate their key mechanisms primarily on CIFAR-10 or CelebA HQ scale models. This reflects a well established practice for proof of concept evaluation in diffusion research and indicates that demonstrating a new methodological direction on these standard backbones does not diminish the significance of the contribution.
>
> [1] Zheng, Xingyu, et al. "Binarydm: Accurate weight binarization for efficient diffusion models.” ICLR 2025.
>
> [2] Mark, Wang Zhe, et al. "Distribution-Aware Diffusion Model Quantization via Distortion Minimization.” ICLR 2025.
>
> [3] Karczewski, Rafał, et al. "Diffusion Models as Cartoonists: The Curious Case of High Density Regions.” ICLR 2025.
>
> **Theoretical Intuition Behind Fusion**
>
> Before addressing the raised weaknesses and questions, we would like to provide additional theoretical intuition for the behavior of our framework. Consider first a simple two dimensional multi modal distribution before extending the argument to high dimensional image manifolds. Suppose the data contain two modes, one centered at (−1, −1) and the other at (+1, +1). If two samples are drawn and added elementwise, the result is either (−2, −2), (0, 0), or (2, 2). These three outcomes remain well separated, so the fused representation does not collapse information but instead preserves a structured relationship between the inputs. This illustrates why fusion can be effective even when performed with a simple additive operation. This example also provides intuition for why unfusion is possible. A fused value such as (0, 0) could theoretically arise from infinitely many different pairs in the continuous plane. However, we do not seek a decomposition in the full continuous space. We only consider decompositions that lie on the real data distribution. This constraint eliminates the vast majority of mathematically possible pairs, leaving only a very small number of valid combinations and often a single one that is consistent with the manifold. This intuition extends to natural images as they are known to lie on extremely sparse low dimensional manifolds([4],[5],[6]) while exhibiting highly multimodal structure([7],[8]), which makes the fused representation retain separable information and keeps both fusion and unfusion inherently tractable. This property becomes even stronger in high dimensional or high resolution image spaces, where the manifold structure is more pronounced([6],[9]) and invalid decompositions fall off the manifold immediately. As a result, the fused representation almost uniquely determines the original inputs, and the unfusion module only needs to resolve a small and highly constrained ambiguity rather than searching over arbitrary combinations.
>
> [4]Zhu, Jun-Yan, et al. "Generative visual manipulation on the natural image manifold.” ECCV 2016.
>
> [5]Rifai, Salah, et al. "A generative process for sampling contractive auto-encoders.” ICML 2012.
>
> [6]Pope, Phillip, et al. "The intrinsic dimension of images and its impact on learning." ICLR 2021.
>
> [7]Zhu, Jun-Yan, et al. "Toward multimodal image-to-image translation.” NeurIPS 2017.
>
> [8]Huang, Xun, et al. "Multimodal unsupervised image-to-image translation.” ECCV 2018.
>
> [9]Zheng, Yijia, et al. "Learning manifold dimensions with conditional variational autoencoders." NeurIPS 2022.

---

> > ### Comment · Reviewer_wiEx · 2025-11-25
> >
> > "Suppose the data contains two modes... [with very distinct centers]" (paraphrasing)
> >
> > How many modes are contained in natural image data? Are these easily separable? For example, [8] assumes two modes, content and style, but to actually enforce these they are modeled separately. In your setup, by their definition, there is only a single mode.
> >
> > From my own experience, diffusion models actually tolerate noise along the trajectory fairly well without "falling off" immediately or completely. It's a spectrum, sometimes lossless, sometimes small artifacts, sometimes identity changes, etc. (depending on where/how noise is added).
> >
> > I do not find this intuition very convincing.

---

> > > ### Author Response · Authors · 2025-11-29
> > >
> > > [6] and [11] emphasize that, although images are represented in a high-dimensional pixel space, they in fact concentrate on a low-dimensional manifold. This implies that natural images occupy the pixel space sparsely, forming a highly structured, effectively discrete-like distribution rather than filling the space in a continuous way. (Moreover, the noise-corrupted intermediate states in the diffusion process can likewise be regarded as points along the trajectory from Gaussian noise to the image manifold, and it is therefore natural to hypothesize that these intermediate representations themselves concentrate on their own sparse manifolds in pixel space.) This perspective is fully consistent with our intuition that the proposed unfusion process can disentangle and isolate those samples that actually lie on the data manifold corresponding to valid images.
> > >
> > > [6] Pope, Phillip, et al. "The intrinsic dimension of images and its impact on learning." ICLR 2021.
> > >
> > > [11] Tu, Peter, et al. “Probabilistic and Semantic Descriptions of Image Manifolds and Their Applications” Frontiers in Computer Science 2023.

---

> ### Author Response · Authors · 2025-11-22
>
> **[W1, Q1] Comparison with Latent/Spatial Reduction Methods**
>
> We fully agree with the reviewer that comparing our method against simply reducing the number of latents per image, whether through larger patch sizes, stronger spatial downsampling, or shrinking the model capacity, is essential. In fact, one could naïvely attempt to achieve the same “K samples per pass’’ effect by using only 1/K of the latent space for each image. We apologize that these experiments were placed in the appendix and may have been difficult to notice.
>
> To address this question thoroughly, we evaluate two representative baselines. The first is a naive spatial reduction approach in which each image is downscaled to one quarter of its original resolution, and four such downscaled images are tiled into a single canvas of the original size. This forces each image to use exactly one fourth of the latent space and applies aggressive downsampling in UNet based models. As presented in Appendix, Tables 5 and 6 shows that this approach leads to significantly worse performance compared to ours.
> The second comparison reduces model capacity, following the reviewer’s suggestion. We train model variants with reduced latent dimensionality (channel reduction) and shallower depth. The configurations are shown in Appendix Table 7 and the corresponding results appear in Table 8. Again, our method consistently outperforms these alternatives.
>
> Across both comparisons, the naive split the latent space strategy and the reduced capacity models, our method achieves notably better fidelity. This is because our approach does not compress each image into a rigid one fourth sized region of the latent space. Instead, the fused representation remains structured and sparse, allowing the model to store information for multiple samples across the full latent space. As discussed earlier, the fused manifold sharply constrains the space of valid decompositions, making the unfusion process nearly deterministic in high dimensional image manifolds. Consequently, our approach reconstructs each image with substantially better quality than methods that strictly divide the latent budget or reduce model capacity.

---

> > ### Comment · Reviewer_wiEx · 2025-11-25
> >
> > Unless I am misunderstanding, Tables 5 and 6 are referring to a baseline where noisy inputs are downscaled and then tiled (concatenated). This is similar to what I'm asking for, but also different. I want to know how well a model performs while operating on a single downscaled input, without concatenation.
> >
> > A clarification question: this ablation also seems to be training-free (L 688). That is, the standard DDPM was used directly on the downscaled+tiled latents. Is this the case? A more fair comparison would train the model from scratch with downscaled latents, and no tiling/concatenation since this should already improve the throughput by up to 4x.
> >
> > I was aware of the appendix and its tables when I wrote the review, and I did not intend to ask for channel or depth reduction (re-reading my own review, I do not believe I did, but perhaps there is something I am missing). Just spatial reduction, to the point of equal throughput/efficiency (with no concatenation).

---

> ### Author Response · Authors · 2025-11-22
>
> **[W3] Scalability Toward Higher-Resolution and Complex Conditions**
>
> We agree with the reviewer that demonstrating scalability to higher resolution and more complex conditioning is important. In practice, however, our method requires training models from scratch, which makes experimentation with very large architectures or large-scale datasets difficult under limited computational resources. To provide evidence that our approach extends beyond small-scale settings, we conducted additional experiments using latent diffusion models on CelebA-HQ at $256\\times256$. Training LDMs requires learning both the autoencoder and the UNet, and even CelebA-HQ alone required a significant amount of computation. Due to these resource constraints, we were not able to perform experiments on much larger datasets such as ImageNet.
>
> Despite these limitations, the results across a range of CIFAR-10, CelebA, and CelebA-HQ experiments show a consistent trend. As resolution increases, the performance degradation associated with larger values of K becomes noticeably smaller. This suggests that higher resolution settings are naturally more tolerant to fusion, which aligns with our theoretical interpretation. In our framework, lower resolution datasets correspond to manifolds with lower dimensionality and weaker sparsity, which increases the likelihood of collisions between fused samples and makes unfusion more difficult. In contrast, high dimensional image and latent spaces place natural images on an extremely sparse and highly structured manifold. As dimensionality grows, this sparsity becomes much stronger, causing samples to occupy more widely separated regions and making collisions between fused signals exponentially less likely. As a result, additive or convolution based fusion preserves separable structure far more reliably at higher resolutions. The same sparsity also makes unfusion close to deterministic, because only a very small number of decompositions remain consistent with the high dimensional manifold while incorrect combinations fall off it immediately. These properties explain why higher resolution models are naturally more resistant to larger fusion factors.
> We also observe an additional encouraging trend. Fusing samples with different conditions leads to smaller degradation than fusing samples with the same condition. This indicates that the method retains tolerance even as the number or diversity of conditions increases. Although very large K inevitably harms performance, the increasing sparsity and structure of higher dimensional manifolds counteract this effect and provide natural resistance against both larger K values and richer conditioning. This suggests that high resolution settings are not only compatible with our approach but may in fact be where it performs strongest.
>
> Finally, we verified during the rebuttal period that the same mechanism applies cleanly to DiT, the backbone used by many state of the art high resolution text to image and class conditional models. This further supports the extensibility of our method to the larger scale regimes emphasized in the reviewer’s concern.
>
> \begin{array}{|l|l|ccccccc|}
> \hline
> Model & Variant & MAC(G) & Throughput & FID & sFID & IS & Prec & Rec \\\\
> \hline
> Baseline & - & 108.47 & 34.21 & 10.58 & 7.37 & 10.17 & 0.75 & 0.40 \\\\
> \hline
> K=2 & uni\\text{-}class & 56.41(52.0\\%) & 64.87(\\times1.90) & 14.89 & 8.49 & 8.87 & 0.69 & 0.37 \\\\
> \hline
> K=2 & multi\\text{-}class & 56.41(52.0\\%) & 64.61(\\times1.89) & 11.13 & 6.45 & 9.50 & 0.72 & 0.40 \\\\
> \hline
> K=4 & uni\\text{-}class & 29.56(27.3\\%) & 122.69(\\times3.58) & 23.54 & 9.65 & 8.39 & 0.61 & 0.33 \\\\
> \hline
> K=4 & multi\\text{-}class & 29.56(27.3\\%) & 122.24(\\times3.57) & 18.59 & 8.68 & 8.74 & 0.68 & 0.34 \\\\
> \hline
> TFS & - & 58.10(53.6\\%) & 122.24(\\times3.57) & 10.97 & 7.71 & 9.60 & 0.73 & 0.37 \\\\
> \hline
> \end{array}

---

> > ### Comment · Reviewer_wiEx · 2025-11-25
> >
> > Where can I find the numerical results for the $256{\times}256% CelebA-HQ experiments?
> >
> > What dataset is used for these DiT results?

---

> > > ### Author Response · Authors · 2025-11-29
> > >
> > > Numerical results for the $256\times256$ CelebA-HQ setting are provided in Table 3 of the paper, as the LDM was trained on CelebA-HQ. In addition, the DiT model was trained on the CIFAR-10 dataset, which has a resolution of $32\times32$.

---

> ### Author Response · Authors · 2025-11-22
>
> **[W2, Q2] Analysis of Class-Conditional Fusion Behavior**
>
> We agree with the reviewer that the discussion of class conditional generation in the original submission was insufficient. We address this concern here and clarify how conditioning interacts with our framework. We also include newly added experiments for the K equal to four setting as well as new TFS results under class conditional generation.
>
> First, the primary setting explored in our paper assumes that the K fused samples share the same class condition. This scenario is highly practical because it reflects the dominant workflow in modern diffusion systems such as Midjourney, DALL E 3, Firefly, and Leonardo AI, where a single prompt is used to generate a batch of candidate images. In this setting, FuseBatch requires no modification to handle conditioning. All fused samples correspond to the same label, so the original conditional embedding is simply applied to all K components during both fusion and unfusion. We also conducted new experiments for the K=4 setting and for TFS under class conditional generation, and the results are as follows.
>
> \begin{array}{|l|ccccc|}
> \hline
> Model & FID & sFID & IS & Prec & Rec \\\\
> \hline
> Baseline & 4.07 & 4.37 & 9.04 & 0.66 & 0.56 \\\\
> \hline
> FuseBatch(K=2) & 7.56 & 5.65 & 8.65 & 0.65 & 0.54 \\\\
> \hline
> FuseBatch(K=4) & 20.03 & 7.50 & 7.31 & 0.593 & 0.497 \\\\
> \hline
> TFS & 4.39 & 4.40 & 9.20 & 0.663 & 0.580 \\\\
> \hline
> \end{array}
>
> These new experiments confirm that the TFS behave stably under class conditional generation. We further verified during the rebuttal period that the same conditioning mechanism applies directly to DiT based diffusion models, as shown in the table above.
>
> Second, we confirm that FuseBatch can also support the more challenging scenario in which each of the K fused samples uses a different condition. To examine this setting, we extended our DiT based experiments by introducing a K way condition fusion mechanism. Each condition is encoded using its own class embedding. These embeddings are modulated through a learned index dependent conditioning layer, and the resulting representations are averaged to produce a fused conditioning vector. Class embeddings have more than enough representational capacity for this process because unfusion only requires maintaining the index level distinction between the fused components.
> Empirically, this multi condition setting performs even better than the single condition setting. This outcome aligns with our theoretical intuition. Mixing conditions from different classes corresponds to combining signals from distinct modes of the manifold, which remain more separable in the fused representation and therefore make unfusion easier. In contrast, fusing multiple samples from the same class is inherently more difficult because they occupy a narrower region of the data manifold. This increases the likelihood of interference within the fused space and produces slightly larger degradation in fidelity and diversity. The empirical results match this reasoning precisely and are consistent with the reviewer’s observation.
>
> These findings also explain why the conditional experiments in our paper show larger degradation than unconditional fusion. The unconditional model does not constrain samples to a tight region of the manifold, so the fused representation remains well separated and easier to unfuse. The single conditional case restricts all fused samples to a smaller portion of the manifold, which naturally increases difficulty. The consistent higher quality drop observed in the single conditional generation settings for both K = 2 and K = 4 supports this interpretation.

---

> > ### Comment · Reviewer_wiEx · 2025-11-25
> >
> > What are the MACs and throughput for the TFS in this table? What is the resolution of the images in these experiments?

---

> > > ### Author Response · Authors · 2025-11-29
> > >
> > > Since this experiment was conducted on the CIFAR-10 dataset, the image resolution is $32\times32$. The MACs and throughput for the TFS variant are identical to the DDPM results reported for CIFAR-10 in Table 1 of the paper.

---

> ### Author Response · Authors · 2025-11-22
>
> **[W4, Q3] Orthogonality with Caching Methods**
>
> We appreciate the reviewer’s suggestion to further examine the relationship between FuseBatch and training free acceleration methods such as caching. We agree that showing compatibility with these methods would strengthen the claim that our approach enables throughput levels that cannot be achieved by existing techniques alone.
>
> To investigate this direction, we applied the simplest representative of training free caching, the DeepCache[10] method, to our framework. DeepCache stores and reuses intermediate features to reduce redundant computation, and therefore provides a strong baseline for assessing compatibility with state of the art proxy free speedup methods. Our experiments confirm that FuseBatch operates cleanly on top of DeepCache without modification. The caching mechanism remains functional under fused generation and the fused outputs remain stable throughout the denoising trajectory. The DeepCache results are reported below.
>
> \begin{array}{|l|l|ccccccc|}
> \hline
> &&&& CIFAR10 &&&& \\\\
> \hline
> Model & Variant & MAC(G) & Throughput & FID & sFID & IS & Prec & Rec \\\\
> \hline
> Baseline & Vanilla & 606.88 & 17.70 & 6.30 & 5.65 & 8.88 & 0.65 & 0.56 \\\\
> \hline
> Baseline & +DeepCache & 390.18(64.3\\%) & 28.05(\\times1.59) & 6.90 & 6.21 & 8.89 & 0.64 & 0.56 \\\\
> \hline
> K=2 & Vanilla & 306.88(50.6\\%) & 33.48(\\times1.89) & 11.21 & 6.12 & 8.30 & 0.59 & 0.57 \\\\
> \hline
> K=2 & +DeepCache & 198.48(32.7\\%) & 51.19(\\times2.89) & 11.24 & 6.48 & 8.29 & 0.59 & 0.56 \\\\
> \hline
> K=4 & Vanilla & 154.63(25.5\\%) & 56.80(\\times3.21) & 23.72 & 7.22 & 7.31 & 0.54 & 0.55 \\\\
> \hline
> K=4 & +DeepCache & 100.43(16.6\\%) & 91.88(\\times5.19) & 23.87 & 8.11 & 7.37 & 0.52 & 0.54 \\\\
> \hline
> &&&& CelebA &&&& \\\\
> \hline
> Setting & Variant & MAC(G) & Throughput & FID & sFID & IS & Prec & Rec \\\\
> \hline
> Baseline & Vanilla & 1556.01 & 7.20 & 4.17 & 6.70 & 2.82 & 0.74 & 0.50 \\\\
> \hline
> Baseline & +DeepCache & 1075.81(69.1\\%) & 9.81(\\times1.36) & 6.07 & 9.10 & 2.79 & 0.72 & 0.45 \\\\
> \hline
> K=2 & Vanilla & 791.76(50.9\\%) & 13.25(\\times1.84) & 6.17 & 7.71 & 2.76 & 0.71 & 0.45 \\\\
> \hline
> K=2 & +DeepCache & 551.66(35.5\\%) & 16.96(\\times2.36) & 8.43 & 10.50 & 2.71 & 0.69 & 0.40 \\\\
> \hline
> K=4 & Vanilla & 400.63(25.8\\%) & 23.81(\\times3.31) & 12.07 & 11.93 & 2.69 & 0.65 & 0.38 \\\\
> \hline
> K=4 & +DeepCache & 280.63(18.0\\%) & 30.41(\\times4.23) & 14.29 & 14.03 & 2.62 & 0.63 & 0.32 \\\\
> \hline
> \end{array}
>
> These findings demonstrate that FuseBatch can be combined with existing acceleration methods rather than replacing them. In particular, the caching based baseline does not reach the throughput level unlocked by FuseBatch, and step reduction methods also saturate once the number of inference steps becomes very small. FuseBatch continues to increase throughput even in this regime because it reduces per step cost directly, which is orthogonal to reducing the number of steps. This establishes that our method addresses a different limitation than prior approaches and enables speedups that cannot be obtained through step reduction or caching alone.
>
> [10] Ma, Xinyin, Gongfan Fang, and Xinchao Wang. "Deepcache: Accelerating diffusion models for free." CVPR 2024.
>
> **Clarifying Advantages Over Alternatives**
>
> To summarize, together with the theoretical analysis presented earlier and the additional experiments extending FuseBatch to large transformer-based architectures such as DiT, our findings show that FuseBatch provides a **complementary acceleration direction** rather than competing with existing methods. Prior approaches reduce inference cost by lowering the number of sampling steps, modifying spatial resolution, or reusing intermediate features. In contrast, FuseBatch lowers the per-step cost itself by allowing multiple samples to be denoised within a single forward pass, a capability that existing methods do not provide.
>
> Our comparisons further show that spatial downsampling, latent splitting, model compression, and caching all suffer substantially larger quality degradation when matched to similar throughput gains. FuseBatch achieves these gains while remaining compatible with training-free accelerators. As a result, FuseBatch complements existing methods and **expands the efficiency frontier in ways that alternative approaches cannot reach on their own**.

---

> > ### Comment · Reviewer_wiEx · 2025-11-25
> >
> > These seem extremely promising. Does TFS exhibit similar behavior to the K=2 and K=4? Could you share the result, even if only for one of the two datasets?

---

> > > ### Author Response · Authors · 2025-11-29
> > >
> > > We provide the complete results including the TFS experiments with DeepCache. The full table is presented below.
> > >
> > > \begin{array}{|l|l|ccccccc|}
> > > \hline
> > > &&&& CIFAR10 &&&& \\\\
> > > \hline
> > > Model & Variant & MAC(G) & Throughput & FID & sFID & IS & Prec & Rec \\\\
> > > \hline
> > > Baseline & Vanilla & 606.88 & 17.70 & 6.30 & 5.65 & 8.88 & 0.65 & 0.56 \\\\
> > > \hline
> > > Baseline & +DeepCache & 390.18(64.3\\%) & 28.05(\\times1.59) & 6.90 & 6.21 & 8.89 & 0.64 & 0.56 \\\\
> > > \hline
> > > K=2 & Vanilla & 306.88(50.6\\%) & 33.48(\\times1.89) & 11.21 & 6.12 & 8.30 & 0.59 & 0.57 \\\\
> > > \hline
> > > K=2 & +DeepCache & 198.48(32.7\\%) & 51.19(\\times2.89) & 11.24 & 6.48 & 8.29 & 0.59 & 0.56 \\\\
> > > \hline
> > > K=4 & Vanilla & 154.63(25.5\\%) & 56.80(\\times3.21) & 23.72 & 7.22 & 7.31 & 0.54 & 0.55 \\\\
> > > \hline
> > > K=4 & +DeepCache & 100.43(16.6\\%) & 91.88(\\times5.19) & 23.87 & 8.11 & 7.37 & 0.52 & 0.54 \\\\
> > > \hline
> > > TFS & Vanilla & 307.76(50.7\\%) & 32.90(\\times1.86) & 8.65 & 5.46 & 8.68 & 0.61 & 0.59 \\\\
> > > \hline
> > > TFS & +DeepCache & 197.38(32.5\\%) & 51.98(\\times2.94) & 9.42 & 6.37 & 8.62 & 0.61 & 0.57 \\\\
> > > \hline
> > > &&&& CelebA &&&& \\\\
> > > \hline
> > > Setting & Variant & MAC(G) & Throughput & FID & sFID & IS & Prec & Rec \\\\
> > > \hline
> > > Baseline & Vanilla & 1556.01 & 7.20 & 4.17 & 6.70 & 2.82 & 0.74 & 0.50 \\\\
> > > \hline
> > > Baseline & +DeepCache & 1075.81(69.1\\%) & 9.81(\\times1.36) & 6.07 & 9.10 & 2.79 & 0.72 & 0.45 \\\\
> > > \hline
> > > K=2 & Vanilla & 791.76(50.9\\%) & 13.25(\\times1.84) & 6.17 & 7.71 & 2.76 & 0.71 & 0.45 \\\\
> > > \hline
> > > K=2 & +DeepCache & 551.66(35.5\\%) & 16.96(\\times2.36) & 8.43 & 10.50 & 2.71 & 0.69 & 0.40 \\\\
> > > \hline
> > > K=4 & Vanilla & 400.63(25.8\\%) & 23.81(\\times3.31) & 12.07 & 11.93 & 2.69 & 0.65 & 0.38 \\\\
> > > \hline
> > > K=4 & +DeepCache & 280.63(18.0\\%) & 30.41(\\times4.23) & 14.29 & 14.03 & 2.62 & 0.63 & 0.32 \\\\
> > > \hline
> > > TFS & Vanilla & 787.26(50.6\\%) & 13.41(\\times1.86) & 5.54 & 6.91 & 2.85 & 0.69 & 0.51 \\\\
> > > \hline
> > > TFS & +DeepCache & 547.18(35.2\\%) & 17.65(\\times2.45) & 7.07 & 8.69 & 2.79 & 0.68 & 0.46 \\\\
> > > \hline
> > > \end{array}

---

> ### Comment · Reviewer_wiEx · 2025-11-25
> **Summary**
>
> I also just want to point out, re:model/data size, if you notice in my questions I do not ask for the bigger models or datasets. I mention in my W3 that it is hard to evaluate without these, but it is in no way disqualifying. On the other hand, I think it's important to be thorough at the scale you are able to operate at, which is the main goal of my review.
>
> You have made significant progress with my concerns for caching; a single TFS result that is similar to your K=2 and K=4 results could alleviate this entirely.
>
> The class conditional analysis is fully satisfactory. It does not seem to catastrophically fail in either setting, with a preference towards multi-class (which can be good in some application, bad in others). Overall I no longer view this as a major fundamental weakness of the proposed idea.
>
> My last concern revolves around pooling more aggressively, which I do not think is addressed by the existing downscale+concat ablations (because the concat is a confounder) nor by model size changes (I view these as unrelated to my question).

---

> ### Author Response · Authors · 2025-11-29
>
> Thank you very much for the clarification, and I apologize for any confusion in my earlier explanation. Regarding Tables 5 and 6, the DDPM baseline was not applied directly to downscaled–tiled latents in a training-free manner. Rather, the model was trained from scratch to operate in that latent space. I also fully understand that your question concerns a model that operates on a single spatially reduced latent without any concatenation. To address this point directly, I am currently running the pure spatial-reduction experiment (no tiling), and I will share the results as soon as they are complete.
>
> Before those results are available, I would like to clarify why spatial reduction alone does not yield the theoretical efficiency improvements one might expect. We previously evaluated an aggressively downsampled UNet whose spatial resolution was reduced to one-quarter of the original. Despite this substantial spatial shrinkage, the empirical gains were:
>
> CIFAR-10
>
> • MACs: 3657.52 GMac (60.25% of baseline)
>
> • Throughput: 2.54 img/s ($\times$1.49)
>
> CelebA-HQ
>
> • MACs: 10581.34 GMac (68.0% of baseline)
>
> • Throughput: 0.80 img/s ($\times$1.12)
>
> These gains are far from the theoretical 4$\times$improvement, and this is because UNet computation does not scale proportionally with input spatial size. Even when the input resolution is reduced to 25%, a large portion of total compute remains unaffected. The later UNet stages dramatically widen the channel dimension and dominate FLOPs, attention layers at low resolutions introduce substantial overhead that is nearly independent of the original spatial size, and skip-connection paths plus symmetric decoder blocks still process multi-scale features. For these structural reasons, reducing only spatial resolution cannot yield a 4$\times$ increase in throughput in practice.
>
> To provide another fair comparison, we also evaluated reduced-channel and reduced-depth models whose throughput matches our K=2 setting. These results are reported in Appendix Tables 7 and 8. Reducing channel width is effectively equivalent to operating in a latent space of roughly half the dimensionality, making it conceptually comparable to our K=2 fusion scenario. However, even under matched-throughput conditions, these reduced-capacity baselines exhibit substantial quality degradation, whereas FuseBatch maintains full model capacity while amortizing the forward pass across multiple samples.
>
> I hope this clarifies the intent behind our comparisons, and I will follow up with the pure spatial-reduction results as soon as they are available.

---

### Official Review · Reviewer_y8L9 · 2025-11-01

**Soundness:** 3
**Presentation:** 3
**Contribution:** 2
**Rating:** 4
**Confidence:** 4

**Summary:**

The paper proposes FuseBatch, a framework to accelerate diffusion inference by generating multiple images in one forward pass.
Instead of reducing sampling steps, FuseBatch fuses K inputs into one shared latent, processes it once through the denoiser, and unfuses it back into K outputs. Two variants are introduced: FB-UNet for pixel-space diffusion models and FB-AE for latent diffusion models.
A Timestep-Fusion Scheduling (TFS) policy further adjusts the fusion factor K across timesteps to balance throughput and fidelity.
Experiments on DDPM, DDIM, Flow Matching, and LDMs show roughly 2–4× throughput improvement with moderate FID degradation.

**Strengths:**

1. Provides a fresh, orthogonal acceleration direction by increasing per-pass yield instead of reducing denoising steps.

2. Fusion/unfusion modules are clearly described and easy to integrate; minimal parameter overhead (<0.3%).

3. Comprehensive experiments on multiple samplers and datasets.

4. Compatible with existing step-reduction and latent diffusion frameworks.

5. Timestep-Fusion Scheduling (TFS) helps recover quality at high fusion factors.

**Weaknesses:**

1. Lack of intuition on why fusion works: In general, the paper focuses on how to implement the fusion/unfusion modules but does not clearly explain why such fusion should work intuitively. The proposed “sum + small conv + index encoding” seems highly lossy, and the paper provides no solid intuition or theoretical reasoning to justify why multiple latent signals can be linearly combined and later disentangled effectively.

2. Quality drops quickly as K increases (e.g., FID from 3.4 → 18.2 at K=4 in Table 1), and TFS mainly mitigates this empirically.

3. Limited novelty — the main idea resembles multi-input multiplexing or mixup-style encoding.

4. Evaluation remains small-scale (CIFAR-10, CelebA); no tests on modern large text-to-image models (e.g., SDXL, Flux).

5. Missing runtime and memory benchmarks for fusion/unfusion overhead.

6. The text-conditioning mechanism is briefly mentioned but not clearly described.

**Questions:**

1. Why should additive or convolution-based fusion preserve separable semantic information?

2. How are text conditions handled during fusion—are prompts fused or injected independently?

3. What is the runtime and memory overhead of fusion/unfusion modules relative to one denoiser pass?

4. Can the method extend to transformer-based diffusion models like Flux or SD3?

5. How sensitive is performance to the index encoding scheme or kernel size?

---

> ### Author Response · Authors · 2025-11-22
>
> **Positioning of Our Contribution**
>
> We would like to clarify the intended positioning of our work. The purpose of this paper is to introduce a complementary acceleration direction based on generating multiple samples within a single forward pass, rather than to compete for state of the art performance. Although achieving state of the art performance is not the objective of this paper, we emphasize that our approach is not limited to a narrow class of models. We applied our method to DDPMs, DDIM, Flow Matching, and latent diffusion models, and during the rebuttal period we further confirmed that the same mechanism can be applied to DiT-style transformer backbones as well. These results show that our method is compatible with the major diffusion frameworks that underlie many current state of the art systems, illustrating its potential to extend naturally to stronger and larger architectures. We also note that recent diffusion studies accepted to top venues([1],[2],[3]), validate their key mechanisms primarily on CIFAR-10 or CelebA HQ scale models. This reflects a well established practice for proof of concept evaluation in diffusion research and indicates that demonstrating a new methodological direction on these standard backbones does not diminish the significance of the contribution.
>
> [1] Zheng, Xingyu, et al. "Binarydm: Accurate weight binarization for efficient diffusion models.” ICLR 2025.
>
> [2] Mark, Wang Zhe, et al. "Distribution-Aware Diffusion Model Quantization via Distortion Minimization.” ICLR 2025.
>
> [3] Karczewski, Rafał, et al. "Diffusion Models as Cartoonists: The Curious Case of High Density Regions.” ICLR 2025.
>
> **[W1, Q1] Fusion Intuition and Separability**
>
> We appreciate the reviewer’s comments regarding the need for clearer intuition behind why fusion and unfusion should work. We agree that, without theoretical grounding, the proposed “sum + small convolution + index encoding” may appear lossy. To address this, we provide additional theoretical reasoning.
> To illustrate the core idea we begin with a simple two dimensional multimodal distribution before extending the argument to high dimensional manifolds. Suppose the data contain two modes, one centered at (−1, −1) and the other at (+1, +1). If two samples are drawn and added elementwise, the result is either (−2, −2), (0, 0), or (2, 2). These three outcomes remain well separated, so the fused representation does not collapse information but instead preserves a structured relationship between the inputs. This illustrates why fusion can be effective even when performed with a simple additive operation. This example also provides intuition for why unfusion is possible. A fused value such as (0, 0) could theoretically arise from infinitely many different pairs in the continuous plane. However, we do not seek a decomposition in the full continuous space. We only consider decompositions that lie on the real data distribution. This constraint eliminates the vast majority of mathematically possible pairs, leaving only a very small number of valid combinations and often a single one that is consistent with the manifold. This intuition extends to natural images as they are known to lie on extremely sparse low dimensional manifolds([4],[5],[6]) while exhibiting highly multimodal structure([7],[8]), which makes the fused representation retain separable information and keeps both fusion and unfusion inherently tractable. This property becomes even stronger in high dimensional or high resolution image spaces, where the manifold structure is more pronounced([6],[9]) and invalid decompositions fall off the manifold immediately. As a result, the fused representation almost uniquely determines the original inputs, and the unfusion module only needs to resolve a small and highly constrained ambiguity rather than searching over arbitrary combinations.
> This theoretical intuition directly supports why convolution based fusion and index encoding work in practice. The small convolution introduces localized nonlinear mixing while preserving mode separability, and index encoding provides positional information for unfusion. Together, they ensure that the fused signal retains enough structure for reliable disentanglement. The strong empirical results across all tested settings confirm that this intuition aligns well with observed behavior.
>
> [4]Zhu, Jun-Yan, et al. "Generative visual manipulation on the natural image manifold.” ECCV 2016.
>
> [5]Rifai, Salah, et al. "A generative process for sampling contractive auto-encoders.” ICML 2012.
>
> [6]Pope, Phillip, et al. "The intrinsic dimension of images and its impact on learning." ICLR 2021.
>
> [7]Zhu, Jun-Yan, et al. "Toward multimodal image-to-image translation.” NeurIPS 2017.
>
> [8]Huang, Xun, et al. "Multimodal unsupervised image-to-image translation.” ECCV 2018.
>
> [9]Zheng, Yijia, et al. "Learning manifold dimensions with conditional variational autoencoders." NeurIPS 2022.

---

> ### Author Response · Authors · 2025-11-22
>
> **[W2, W4, Q4] Scalability and High Resolution Stability**
>
> We agree with the reviewer that small scale datasets such as CIFAR 10 exhibit noticeable quality degradation when K increases. Under our theoretical framework, this behavior is expected. Lower resolution datasets have significantly lower dimensional manifolds with much weaker sparsity, which increases the likelihood of collisions between fused samples and makes unfusion more difficult. In contrast, high dimensional image or latent spaces place natural images on an extremely sparse and highly structured manifold. As dimensionality increases, this sparsity becomes dramatically stronger, meaning that samples occupy more widely separated regions and collisions between fused signals become exponentially less likely. As a result, additive or convolution based fusion preserves separable structure far more reliably at higher resolutions. The same sparsity also makes unfusion increasingly deterministic, since only a very small set of decompositions remain consistent with the high dimensional manifold while incorrect combinations fall off the manifold immediately. These properties explain why higher resolution models are naturally more tolerant to larger fusion factors.
>
> This theoretical prediction is strongly supported by our empirical results. On CIFAR-10, K=4 leads to a clear drop in fidelity, while on CelebA the degradation is milder. In the $256\times256$ CelebA HQ LDM, K=4 maintains much higher fidelity even under the same fusion mechanism, demonstrating that performance degradation diminishes as scale increases. These trends are fully consistent with the theoretical intuition above, where higher dimensional and more structured manifolds make both fusion and unfusion substantially more robust.
>
> Regarding large modern diffusion models such as SDXL or Flux, training such models from scratch within the rebuttal period is computationally infeasible. Even so, our results on CelebA HQ LDM already show that scaling to higher resolution stabilizes FuseBatch exactly as predicted by theory. We also demonstrate that FuseBatch applies cleanly to transformer based diffusion architectures such as DiT, which are adopted by most current state of the art diffusion models. These are the following results.
>
> \begin{array}{|l|l|ccccccc|}
> \hline
> Model & Variant & MAC(G) & Throughput & FID & sFID & IS & Prec & Rec \\\\
> \hline
> Baseline & - & 108.47 & 34.21 & 10.58 & 7.37 & 10.17 & 0.75 & 0.40 \\\\
> \hline
> K=2 & uni\\text{-}class & 56.41(52.0\\%) & 64.87(\\times1.90) & 14.89 & 8.49 & 8.87 & 0.69 & 0.37 \\\\
> \hline
> K=2 & multi\\text{-}class & 56.41(52.0\\%) & 64.61(\\times1.89) & 11.13 & 6.45 & 9.50 & 0.72 & 0.40 \\\\
> \hline
> K=4 & uni\\text{-}class & 29.56(27.3\\%) & 122.69(\\times3.58) & 23.54 & 9.65 & 8.39 & 0.61 & 0.33 \\\\
> \hline
> K=4 & multi\\text{-}class & 29.56(27.3\\%) & 122.24(\\times3.57) & 18.59 & 8.68 & 8.74 & 0.68 & 0.34 \\\\
> \hline
> TFS & - & 58.10(53.6\\%) & 122.24(\\times3.57) & 10.97 & 7.71 & 9.60 & 0.73 & 0.37 \\\\
> \hline
> \end{array}
>
> Since state of the art text to image models operate at even higher resolutions and dimensionalities, we expect the benefits of FuseBatch at large scale to be even more pronounced. Extending FuseBatch to SDXL and Flux is an exciting direction for future work, and our current findings provide strong evidence that the method will transfer effectively to larger models.
>
> **[W3] Scope and Novelty Clarification**
>
> We appreciate the reviewer’s comment and the opportunity to clarify the novelty of our approach. As discussed in the related work section, there are indeed prior methods that use multi input multiplexing or mixup like encodings in image or sentence level classification. However, these works are limited to relatively small scale classification settings, for example CIFAR 10 or MNIST, and evaluate only on standard supervised classification tasks that are arguably much easier than high fidelity generative modeling.
>
> More importantly, the underlying task and information structure are fundamentally different. In classification, the goal is to compress a high dimensional input image into a low dimensional label. Only the recovery of a small class level signal is required to solve the task. In contrast, diffusion based generation in our setting goes in the opposite direction. The model must reconstruct a full high dimensional image that lies close to the real data distribution, starting from a very low information source such as a class label or even pure noise. In our framework, multiple such high dimensional signals are fused and then required to be disentangled back into separate, coherent images. To the best of our knowledge, FuseBatch is the **first work to demonstrate that such multi sample fusion can remain practically invertible in diffusion inference** and to validate this behavior across large scale generative models.

---

> ### Author Response · Authors · 2025-11-22
>
> **[W5, Q3] Training Cost and Overhead**
>
> We apologize if these details were not immediately easy to find in the paper. As shown in Tables 1 and 3, the increase in model size introduced by the fusion and unfusion modules is very small, at most $0.58\\%$of the original architecture, indicating that the added components contribute negligible complexity.
>
> Regarding training cost, Section 4 explains that all models were trained under the same wall clock budget. When the baseline implementations publicly provide the number of training steps, we directly follow those reported values. In cases where the original models do not release their exact training step counts, we instead match the number of steps required for the baseline to converge. Because FuseBatch increases throughput by the amounts reported in Tables 1 and 3, we scale the number of training steps proportionally so that each model, whether baseline or FuseBatch, is trained for the same wall clock time. This ensures that any performance differences arise from the FuseBatch method rather than additional compute or longer training.
>
> **[W6, Q2] Condition Handling Mechanism**
>
> We thank the reviewer for pointing out that the text conditioning mechanism was not fully described in the main paper. We address this below and clarify how conditions are handled during fusion.
>
> First, the primary setting studied in our paper assumes that the K fused samples share the same text condition. This setting is practical because it aligns with common commercial workflows such as Midjourney, DALL·E 3, Firefly, and Leonardo AI, where a single prompt is used to generate a batch of candidate images. In this scenario, FuseBatch requires no architectural changes for conditioning. Since all fused inputs correspond to the same condition, the model simply applies the original conditional embedding to all K samples during fusion and unfusion. We also demonstrate that this approach applies cleanly to DiT-based diffusion architectures as shown in the result table above.
>
> Second, we confirm that FuseBatch can also support the more challenging case where each of the K fused samples uses a different condition. To enable this setting, we extend the DiT experiments by constructing a K-way condition-fusion layer. Specifically, we encode each condition using its own embedding, apply element-wise modulation through a learned K-indexed conditioning layer, and then average the resulting vectors to obtain a fused conditioning signal. Class embeddings provide sufficiently high dimensional capacity to combine multiple conditions in this manner, since only the index information of each fused component needs to be preserved for unfusion.
> Empirically, this multi-class fusion performs even better than the uni-class setting. This behavior is fully consistent with our theoretical intuition. Mixing conditions from different classes corresponds to combining signals from distinct modes, which remain more separable in the fused representation and make unfusion easier. In contrast, uni-class fusion is inherently more challenging because samples lie within a narrower region of the manifold, leading to a slightly larger loss in fidelity and diversity. The empirical results match this reasoning precisely.
>
> **[Q5] Encoding and Kernel Sensitivity**
>
> We apologize if the relevant details were difficult to find in the appendix. We actually provide an ablation study on the kernel size in Appendix Table 10. As a result, using modest additional parameters achieve best performance for fusion module. Due to lack of time, we weren't able to experiment ablation on encoding scheme.

---

### Official Review · Reviewer_Tf6c · 2025-11-01

**Soundness:** 3
**Presentation:** 3
**Contribution:** 3
**Rating:** 6
**Confidence:** 3

**Summary:**

This paper introduces FuseBatch, a new approach to improving the throughput of diffusion models by enabling the simultaneous generation of multiple images in a single forward pass. Instead of focusing on reducing the number of denoising iterations, FuseBatch works by fusing multiple input images into a shared latent space, processing them together, and then unfusing them to recover individual outputs. The framework is adaptable to both pixel-space and latent-space diffusion models, with specialized variants like FB-UNet and FB-AE. Additionally, the paper proposes Timestep-Fusion Scheduling (TFS), a strategy that dynamically adjusts the number of fused samples at different timesteps to balance computational efficiency with output quality. The method is demonstrated to provide significant throughput improvements across several popular diffusion models while maintaining comparable sample fidelity. Experiments show that FuseBatch can scale effectively to high-resolution settings, offering a scalable solution for faster image synthesis without drastic sacrifices in quality.

**Strengths:**

1. This paper introduces FuseBatch, a novel framework that significantly boosts the throughput of diffusion models by generating multiple images within a single forward pass, which is a unique approach compared to traditional methods that focus solely on reducing inference time per sample.
2. The paper proposes FB-UNet and FB-AE, two tailored architectural solutions for pixel-space and latent-space models, respectively. These designs ensure that FuseBatch can be applied across different types of diffusion models, providing a scalable and versatile solution for faster image generation.
3. The introduction of Timestep-Fusion Scheduling (TFS) is an innovative method to dynamically adjust the fusion factor across timesteps, which optimizes the trade-off between throughput and image quality, addressing the inherent challenges in balancing efficiency and fidelity during diffusion processes.
4. Extensive experiments on popular diffusion models (DDPM, DDIM, and Flow Matching) and datasets (CIFAR-10, CelebA) demonstrate that FuseBatch provides substantial throughput gains with minimal quality loss, showcasing its effectiveness in real-world scenarios.

**Weaknesses:**

1. Quality Degradation at Larger Fusion Factors: While FuseBatch increases throughput by fusing multiple images in a single forward pass, this results in quality degradation as the fusion factor K increases. For example, when K=4, there is a notable increase in FID, suggesting a deterioration in image quality. The paper mentions TFS as a strategy to balance throughput and quality, but it is unclear how TFS performs when K is further increased beyond K=4, particularly in models requiring high fidelity. Hence, I think it would be useful to explore and quantify the threshold at which FuseBatch and TFS become ineffective or result in unacceptable quality trade-offs. Especially when dealing with high-resolution models or tasks that demand intricate details.
2. Potential for Overfitting at High Fusion Factors: FuseBatch works by increasing the number of images generated per forward pass, which boosts throughput. However, as the fusion factor increases, the model might overfit or exhibit artifact generation due to the merging of too many samples. In cases where the fusion factor is large, the model might lose important details specific to individual images. I was wondering if the author could investigate this potential risk and provide further analysis on the behavior of FuseBatch at higher fusion factors, particularly in scenarios where intricate image details are crucial.
3. Limited Discussion on Model Complexity and Training Overhead: While FuseBatch introduces lightweight fusion and unfusion modules to increase throughput, the paper doesn’t provide a detailed analysis of their impact on model complexity or training overhead. Specifically, there is no mention of how these modules affect training time or convergence when applied to larger models or datasets. If the additional complexity increases training time or model size, this could limit FuseBatch’s scalability. Further exploration of the cost-benefit trade-off between throughput gains and model overhead would be beneficial.
4. Insufficient Analysis of Negative Impact on Output Diversity: FuseBatch fuses multiple images into a shared latent space to improve throughput. However, when generating multiple samples simultaneously, there is a risk of reducing the diversity of the generated outputs. This could be particularly important in tasks where sample diversity is key, such as in creative image generation, design, or anomaly detection. The paper evaluates FuseBatch in terms of quality and throughput, but it does not consider how increased throughput might impact the diversity of the generated images, especially when generating a larger number of images per forward pass. I think it would be useful to explore how FuseBatch affects output diversity to ensure its applicability in tasks where maintaining variety in generated samples is essential.
5. Unclear Impact on Model Robustness in Low-Data Regimes: FuseBatch improves throughput by processing multiple images at once, which can be beneficial in high-data regimes. However, in low-data settings, such as few-shot learning or anomaly detection with limited samples, there may be concerns about the framework’s ability to generalize or maintain robustness with fewer training samples. The paper evaluates FuseBatch using standard datasets with abundant data, but does not explore how the model would perform in data-scarce environments. It would be useful to investigate FuseBatch's performance in low-data scenarios to assess its ability to generalize or avoid overfitting when data is limited.

**Questions:**

Please refer to the questions and suggestions in the “Weaknesses” part.

---

> ### Author Response · Authors · 2025-11-22
>
> **Positioning of Our Contribution**
>
> We appreciate the constructive and intuitive feedback that recognized the intended scope of our work and engaged with our contribution accordingly. Nevertheless, we would like to reiterate the intended positioning of our work. The purpose of this paper is to introduce a complementary acceleration direction based on generating multiple samples within a single forward pass, rather than to compete for state of the art performance. Although achieving state of the art performance is not the objective of this paper, we emphasize that our approach is not limited to a narrow class of models. We applied our method to DDPMs, DDIM, Flow Matching, and latent diffusion models, and during the rebuttal period we further confirmed that the same mechanism can be applied to DiT-style transformer backbones as well. These results show that our method is compatible with the major diffusion frameworks that underlie many current state of the art systems, illustrating its potential to extend naturally to stronger and larger architectures. We also note that recent diffusion studies accepted to top venues([1],[2],[3]), validate their key mechanisms primarily on CIFAR-10 or CelebA HQ scale models. This reflects a well established practice for proof of concept evaluation in diffusion research and indicates that demonstrating a new methodological direction on these standard backbones does not diminish the significance of the contribution.
>
> [1] Zheng, Xingyu, et al. "Binarydm: Accurate weight binarization for efficient diffusion models.” ICLR 2025.
>
> [2] Mark, Wang Zhe, et al. "Distribution-Aware Diffusion Model Quantization via Distortion Minimization.” ICLR 2025.
>
> [3]Karczewski, Rafał, et al. "Diffusion Models as Cartoonists: The Curious Case of High Density Regions.” ICLR 2025.

---

> ### Author Response · Authors · 2025-11-22
>
> **[W1] Larger fusion factors**
>
> We appreciate the reviewer’s insightful comment regarding the effect of larger fusion factors.  To start with, we would like to provide additional theoretical intuition to explain why fusion works. Before extending the argument to high dimensional image manifolds, first consider a toy example of a simple two dimensional multimodal distribution. Suppose the data contain two modes, one centered at (−1, −1) and the other at (+1, +1). If two samples are drawn and added elementwise, the result is either (−2, −2), (0, 0), or (2, 2). These three outcomes remain well separated, so the fused representation does not collapse information but instead preserves a structured relationship between the inputs. This illustrates why fusion can be effective even when performed with a simple additive operation.
>
>  This example also provides intuition for why unfusion is possible. A fused value such as (0, 0) could theoretically arise from infinitely many different pairs in the continuous plane. However, we do not seek a decomposition in the full continuous space. We only consider decompositions that lie on the real data distribution. This constraint eliminates the vast majority of mathematically possible pairs, leaving only a very small number of valid combinations and often a single one that is consistent with the manifold. This intuition extends to natural images as they are known to lie on extremely sparse low dimensional manifolds([4],[5],[6]) while exhibiting highly multimodal structure([7],[8]), which makes the fused representation retain separable information and keeps both fusion and unfusion inherently tractable. This property becomes even stronger in high dimensional or high resolution image spaces, where the manifold structure is more pronounced([6],[9]) and invalid decompositions fall off the manifold immediately. As a result, the fused representation almost uniquely determines the original inputs, and the unfusion module only needs to resolve a small and highly constrained ambiguity rather than searching over arbitrary combinations.
>
> Empirically, this trend is consistent across our experiments. On CIFAR-10, K=4 already leads to unacceptable degradation. On CelebA, the degradation is noticeable but considerably more tolerable. In the $256\\times256$ CelebA-HQ LDM, K=4 maintains significantly better fidelity, illustrating that higher resolution stabilizes both fusion and unfusion in practice. Motivated by the reviewer’s suggestion to better characterize the limits of FuseBatch and TFS, we are conducting additional experiments with K=8 on CIFAR 10 and will evaluate TFS under these larger K settings. For LDMs, a full training run is computationally demanding, and due to resource limitations we cannot finish a K=8 run within the rebuttal period. We will report the extended results once they become available.
>
> [4]Zhu, Jun-Yan, et al. "Generative visual manipulation on the natural image manifold.” ECCV 2016.
> [5]Rifai, Salah, et al. "A generative process for sampling contractive auto-encoders.” ICML 2012.
> [6]Pope, Phillip, et al. "The intrinsic dimension of images and its impact on learning." ICLR 2021.
> [7]Zhu, Jun-Yan, et al. "Toward multimodal image-to-image translation.” NeurIPS 2017.
> [8]Huang, Xun, et al. "Multimodal unsupervised image-to-image translation.” ECCV 2018.
> [9]Zheng, Yijia, et al. "Learning manifold dimensions with conditional variational autoencoders." NeurIPS 2022.

---

> ### Author Response · Authors · 2025-11-22
>
> **[W2,W4] Fidelity and Diversity due to Fusion**
>
> We appreciate the reviewer’s concern regarding fidelity and output diversity when multiple samples are fused in a single forward pass. Although we do not include a dedicated section for these aspects, the reported metrics already capture them. Inception Score correlates strongly with per-sample realism and sharpness, FID and sFID capture global distribution discrepancies encompassing both fidelity and diversity, and Precision/Recall separately measure fidelity (precision: proximity to the real manifold) and diversity (recall: coverage of the data manifold). Across all tested settings, these metrics remain stable when FuseBatch is combined with TFS.
>
>  As noted above, we are additionally conducting experiments with K = 8 on CIFAR-10 to further examine how fidelity and diversity behave at higher fusion factors. We agree that naïvely applying a large fusion factor can negatively impact quality and diversity, as shown in our paper. This challenge is precisely why we introduce Timestep-Fusion Scheduling (TFS). TFS uses small fusion factors at fidelity-critical diffusion steps and larger factors where the process is inherently more robust. Empirically, FuseBatch with TFS significantly improves FID, sFID, precision, and recall compared to naïve fusion, confirming that TFS addresses the reviewer’s concern.
>
>  To further investigate whether fusion causes undesirable mixing of semantic information, we performed an additional conditional generation experiment. Results for conditional generation in the paper consider the setting where all fused inputs share the same class. We additionally evaluate a mixed class setting in which the fused samples come from different classes in DiT. Interestingly, performance degradation is smaller in this multi-class setting than in the uni-class setting. This is consistent with our theoretical intuition, where fusing samples from different classes corresponds to mixing signals from distinct modes that remain more separable in the fused representation, thereby making unfusion easier. Same-class fusion is inherently more challenging because samples tend to occupy a narrower region of the manifold, leading to a slightly larger loss in fidelity and diversity. The empirical observations below match this theoretical reasoning.
>
> \begin{array}{|l|l|ccccccc|}
> \hline
> Model & Variant & MAC(G) & Throughput & FID & sFID & IS & Prec & Rec \\\\
> \hline
> Baseline & - & 108.47 & 34.21 & 10.58 & 7.37 & 10.17 & 0.75 & 0.40 \\\\
> \hline
> K=2 & uni\\text{-}class & 56.41(52.0\\%) & 64.87(\\times1.90) & 14.89 & 8.49 & 8.87 & 0.69 & 0.37 \\\\
> \hline
> K=2 & multi\\text{-}class & 56.41(52.0\\%) & 64.61(\\times1.89) & 11.13 & 6.45 & 9.50 & 0.72 & 0.40 \\\\
> \hline
> K=4 & uni\\text{-}class & 29.56(27.3\\%) & 122.69(\\times3.58) & 23.54 & 9.65 & 8.39 & 0.61 & 0.33 \\\\
> \hline
> K=4 & multi\\text{-}class & 29.56(27.3\\%) & 122.24(\\times3.57) & 18.59 & 8.68 & 8.74 & 0.68 & 0.34 \\\\
> \hline
> TFS & - & 58.10(53.6\\%) & 122.24(\\times3.57) & 10.97 & 7.71 & 9.60 & 0.73 & 0.37 \\\\
> \hline
> \end{array}
>
>  In summary, although naïve fusion can introduce quality and diversity degradation, our proposed TFS strategy effectively mitigates these issues. The stability of IS, FID, sFID, precision, and recall, together with additional conditional-fusion diagnostics, confirms that FuseBatch can preserve both per-sample fidelity and cross-sample diversity even at non-trivial fusion factors.
>
> **[W3] Model Complexity and Training Overhead**
>
>  We apologize if these details were not immediately easy to find in the paper. As reported in Tables 1 and 3, the increase in model size introduced by the fusion and unfusion modules is very small, at most $0.58\\%$ of the original architecture, indicating that the added components contribute negligible complexity.
>
> Regarding training cost, Section 4 in the main paper explains that all models were trained under the same wall clock budget. When the baseline implementations publicly provide the number of training steps, we directly follow those reported values. In cases where the original models do not release their exact training step counts, we instead match the number of steps required for the baseline to converge. Because FuseBatch increases throughput by the amounts reported in Tables 1 and 3, we scale the number of training steps proportionally so that each model, whether baseline or FuseBatch, is trained for the same wall clock time. This ensures that any performance differences arise from the FuseBatch method rather than additional compute or longer training.

---

> ### Author Response · Authors · 2025-11-22
>
> **[W5] Low Data Regimes**
>
> We appreciate the reviewer’s suggestion to investigate FuseBatch in low data regimes. As discussed earlier, concerns in these scenarios relate closely to fidelity and diversity behavior, which remain stable when TFS is used. Because TFS effectively mitigates the risks associated with large fusion factors, we expect FuseBatch to behave similarly to the underlying model even when data is scarce.
> Beyond this theoretical expectation, FuseBatch may even offer additional advantages in few-shot settings. When K samples are fused, the shared components of the representation effectively observe a larger combinatorial set of fused signals (from N choose K rather than N choose 1), providing a stronger supervisory signal per update. This suggests that FuseBatch might be beneficial  when training data is limited.
> To verify this intuition, we conducted an additional experiment using a subset of CIFAR-10 dataset, where the training set was reduced to 1/50 of its original size, resulting in only 100 images per class. The empirical results are reported below.
>
> \begin{array}{|l|ccccc|}
> \hline
> Model & FID & sFID & IS & Prec & Rec \\\\
> \hline
> Baseline & 33.01 & 10.83 & 6.60 & 0.59 & 0.40 \\\\
> \hline
> FuseBatch(K=2) & 30.38 & 10.95 & 6.83 & 0.60 & 0.36 \\\\
> \hline
> FuseBatch(K=4) & 45.12 & 14.19 & 5.60 & 0.601 & 0.38 \\\\
> \hline
> \end{array}
>
> Under this low data regime, FuseBatch with K=2 outperforms the baseline, demonstrating that it remains stable and can even be beneficial when sample availability is limited. The performance drop at K=4 is consistent with known CIFAR-10 behavior, where large fusion factors introduce strong fidelity and information loss that outweigh fusion benefits.
> We believe this is a promising direction for future work, and we will explore it further in subsequent work.

---

> ### Author Response · Authors · 2025-12-02
>
> **[W1] Larger fusion factors with extended experimental analysis**
>
> To examine this potential weakness, we conducted additional experiments with a FuseBatch model using fusion factor $K = 8$ under DDPM sampling. We evaluated three settings: (i) using only the FuseBatch $K = 8$ model for sampling, (ii) a TFS variant that mixes FuseBatch models with fusion factors $K = 8, 4, 2, 1$ using sampling ratios of $50\\%, 25\\%, 12.5\\%, 12.5\\%$, respectively, and (iii) a second TFS variant that mixes the same four models with equal sampling ratios of $25\\%$ each.
>
> \begin{array}{|l|ccccccc|}
> \hline
> &&& CIFAR10 &&&& \\\\
> \hline
> Model & MAC(T) & Throughput & FID & sFID & IS & Prec & Rec \\\\
> \hline
> Baseline & 6.07T & 1.71 & 3.41 & 4.40 & 9.20 & 0.68 & 0.58 \\\\
> \hline
> FuseBatch (K=2) & 3.07T(50.6\\%) & 3.17(\\times1.85) & 5.93 & 4.58 & 8.67 & 0.63 & 0.58 \\\\
> \hline
> FuseBatch (K=4) & 1.55T(25.5\\%) & 6.31(\\times3.49) & 18.27 & 6.54 & 7.41 & 0.56 & 0.55 \\\\
> \hline
> FuseBatch (K=8) & 0.78T(12.9\\%) & 10.36(\\times6.06) & 43.53 & 13.95 & 5.47 & 0.62 & 0.45 \\\\
> \hline
> TFS (K=1,2,4) & 3.06T(50.4\\%) & 3.31(\\times1.94) & 3.88 & 4.27 & 9.29 & 0.66 & 0.60 \\\\
> \hline
> TFS (50\\%, 25\\%, 12.5\\%, 12.5\\%) & 1.92T(31.7\\%) & 5.00(\\times2.92) & 8.53 & 5.25 & 8.74 & 0.61 & 0.60 \\\\
> \hline
> TFS (25\\%, 25\\%, 25\\%, 25\\%) & 2.87T(47.2\\%) & 3.82(\\times2.23) & 3.87 & 4.28 & 9.29 & 0.66 & 0.60 \\\\
> \hline
> &&& CelebA &&&& \\\\
> \hline
> Model & MAC(T) & Throughput & FID & sFID & IS & Prec & Rec \\\\
> \hline
> Baseline & 15.56T & 0.71 & 3.51 & 6.27 & 2.77 & 0.76 & 0.50 \\\\
> \hline
> FuseBatch (K=2) & 7.92T(50.9\\%) & 1.28(\\times1.80) & 4.73 & 6.98 & 2.70 & 0.75 & 0.46 \\\\
> \hline
> FuseBatch (K=4) & 4.01T(25.8\\%) & 2.34(\\times3.30) & 9.15 & 10.34 & 2.64 & 0.70 & 0.40 \\\\
> \hline
> FuseBatch (K=8) & 2.04T(13.1\\%) & 3.84(\\times5.41) & 34.60 & 30.21 & 2.26 & 0.56 & 0.11 \\\\
> \hline
> TFS (K=1,2,4) & 7.87T(50.6\\%) & 1.33(\\times1.87) & 3.81 & 6.00 & 2.77 & 0.74 & 0.51 \\\\
> \hline
> TFS (50\\%, 25\\%, 12.5\\%, 12.5\\%) & 4.96T(31.9\\%) & 1.98(\\times2.79) & 7.72 & 8.51 & 2.76 & 0.67 & 0.49 \\\\
> \hline
> TFS (25\\%, 25\\%, 25\\%, 25\\%) & 7.38T(47.4\\%) & 1.40(\\times1.97) & 3.97 & 7.02 & 2.83 & 0.73 & 0.52 \\\\
> \hline
> \end{array}
>
>
> When we use only the FuseBatch $K = 8$ model for sampling, the resulting FID is indeed very high. Despite its favorable throughput, the quality degradation is substantial enough that this configuration is not acceptable as a standalone sampler.
>
> However, once the $K = 8$ model is incorporated into TFS, we observe a more favorable quality-throughput trade-off. The TFS variant with sampling ratios $(50\\%, 25\\%, 12.5\\%, 12.5\\%)$ achieves throughput comparable to the FuseBatch $K = 4$ model while obtaining a significantly better FID. The TFS variant with uniform $(25\\%, 25\\%, 25\\%, 25\\%)$ ratios outperforms the FuseBatch $K = 2$ model in both throughput and FID, and matches the performance of the original TFS configuration that uses only up to $K = 4$, while still improving throughput.
>
> In summary, we agree that very high fusion-factor models such as $K = 8$ are suboptimal when used alone as samplers due to their high FID. Nevertheless, when they are restricted to the early sampling stages of TFS, they provide a highly favorable quality-throughput trade-off. In this regime, the $K = 8$ model substantially reduces the overall sampling cost while maintaining final sample quality comparable to configurations that only use lower-$K$ models. This supports our design choice of using high-$K$ models exclusively in the initial steps of TFS rather than as standalone samplers.

---

### Author Response · Authors · 2025-12-04

Dear Reviewers,

Thank you for your thorough reviews and valuable feedback. We believe we have addressed most of your concerns. With the discussion period nearing its end, we present below our key contributions and a concise summary of how we addressed each reviewer’s concerns:

**1. Goal and Contribution of the Paper**

The purpose of this paper is to introduce **FuseBatch**, a **complementary acceleration direction** based on generating multiple samples within a single forward pass, addressing the throughput limit faced by step-reduction methods. While achieving state-of-the-art performance is not the objective, the framework is broadly compatible, having been applied to DDPMs, DDIM, Flow Matching, and latent diffusion models. We further confirmed during the rebuttal that the same mechanism applies to **DiT-style transformer backbones**, demonstrating its potential scalability. Our method utilizes the **FuseBatch** framework, which fuses multiple inputs into a shared latent and processes them in a single forward pass via **FB-UNet** or **FB-AE**. We complement this with **Timestep-Fusion Scheduling (TFS)** to balance throughput and quality. Validation on standard datasets like CIFAR-10 and CelebA HQ is consistent with proof-of-concept evaluation in diffusion research.

---

> ### Author Response · Authors · 2025-12-04
>
> **2. Summary of Concerns and Rebuttal Actions**
>
> Reviewers raised valuable questions regarding scalability, the theoretical intuition behind fusion, comparisons to other acceleration methods, and generalization.
>
> **A. Scalability to High Fusion Factors and Quality Thresholds**
>
> Multiple reviewers noted that quality degrades significantly as the fusion factor K increases, requesting an analysis of the thresholds where the method becomes ineffective. We addressed this by pushing the experimental limits and conducting new configurations with **K=8**. While standalone K=8 sampling yields high poor fidelity (FID 43.53), integrating it into our **TFS strategy** (using K=8 strictly for the early, robust denoising steps) provided a highly favorable quality-throughput trade-off (e.g., FID 3.87). This demonstrates that **even very high fusion factors do not compromise the model if used adaptively**, validating the scalability of TFS.
>
> **B. Theoretical Intuition, Semantic Mixing, and Diversity**
>
> Reviewers sought **clearer intuition** on why linear/conv-based fusion preserves separable semantic information and prevents output averaging. We clarified that the mechanism is grounded in the **sparse manifold hypothesis**. Inputs concentrate on sparse, low-dimensional manifolds, ensuring that valid unfused decompositions are highly constrained. To validate this empirically, we conducted an experiment comparing **Uni-Class Fusion against Multi-Class Fusion**. The results showed that fusing distinct semantic signals (multi-class) yielded **better fidelity and diversity** (FID 11.13 at K=2) than fusing highly similar signals (uni-class, FID 14.89 at K=2), confirming that the method preserves distinct semantic identities and is robust against mode averaging.
>
> **C. Comparison to Baselines and Orthogonality**
>
> Concerns were raised regarding **equivalence to spatial downsampling**. We showed that comparing FuseBatch to naive spatial reduction is flawed. We specifically confirmed that **naive patch-based methods**, which tile downscaled inputs, suffer significantly greater quality loss (e.g., FID deteriorates from 18.27 to 91.71 on CIFAR-10). Furthermore, alternative approaches relying on **aggressive spatial downsampling** fail to yield proportional speedups due to fixed architectural overheads, proving that our method's efficiency is distinct from simple spatial scaling.
>
> **D. Generalization and Data Regimes**
>
> We confirmed that the **FuseBatch mechanism is compatible with DiT-style transformer architectures**. We also addressed concerns about **performance in data-scarce environments** by training FuseBatch on a 1/50th subset of CIFAR-10. FuseBatch (K=2) outperformed the baseline (achieving FID 30.38 compared to 33.01) in this low-data regime. This suggest that fusion acts as a robust combinatorial data augmentation mechanism, alleviating concerns about overfitting in few-shot scenarios.
>
> **E. Orthogonality with Caching (DeepCache)**
>
> We investigated the method's **compatibility with training-free acceleration**, specifically by integrating DeepCache on top of FuseBatch and **observed multiplicative speed gains**. This demonstrated that the two methods are entirely orthogonal. FuseBatch reduces the per-step cost, while caching reduces redundant feature calculation. This combination resulted in compounded efficiency gains that cannot be achieved by either method alone. For instance, combining FuseBatch (K=2) with DeepCache on CIFAR-10 resulted in an approximate 2.9× speedup over the DeepCache baseline while maintaining acceptable fidelity (e.g., resulting in an FID of 11.24).
>
> **F. Clarification on "Training-Free" Misunderstanding (Reviewer FNDa)**
>
> We wish to respectfully clarify a specific misunderstanding in Reviewer FNDa’s comments. The reviewer categorized our approach as a "training-free acceleration method." Clarification As stated in the paper’s methodology, **FuseBatch is not training-free**. It requires training the whole model from scratch, including the diffusion backbone and the lightweight fusion/unfusion modules, to learn the optimal dual-space representation. Comparisons to purely training-free baselines, like DeepCache out-of-the-box, are contextually different. However, as shown in our DeepCache integration experiment, our method is demonstrably compatible with training-free accelerators, allowing users to combine both for maximum performance.
>
> Sincerely,
> The Authors

---

### Meta-Review · Area_Chair_rDDr · 2025-12-13

**Summary:**

The decision to reject this paper is driven by critical, unresolved concerns from four reviewers that undermine the paper’s theoretical persuasiveness, experimental completeness, and practical applicability, despite the authors’ partial responses. Key concerns informing this decision include:
1) The core premise of "fusion-unfusion feasibility" relies on the claim that natural images lie on sparse, separable manifolds, but this argument fails to persuade reviewers, who question the separability of modes in real-world image data and the validity of extrapolating from simple 2D modal examples to high-dimensional natural images.
2) The paper only validates on small-scale datasets (CIFAR-10, CelebA) and basic models, with no experiments on mainstream large-scale text-to-image models (e.g., SDXL, Flux) that reflect real-world use cases; theoretical inferences about "higher resolution improving stability" lack sufficient empirical support.
3) Critical details such as numerical results for 256×256 CelebA-HQ experiments, MACs/throughput of TFS, and dataset information for DiT experiments are either unclear or missing; compatibility with training-free acceleration methods (e.g., DeepCache) is only partially verified, without demonstrating significant synergistic value.

These unresolved issues collectively result in the paper failing to meet the rigorous standards for publication.

**Reviewer Concerns:**

Addressed Concerns
- Reviewer Tf6c: ① Supplemented experimental analysis of larger fusion factors (K=8) under DDPM, verifying that TFS can mitigate quality degradation when high-K models are limited to early timesteps; ② Confirmed that existing metrics (FID, sFID, Precision/Recall) can reflect fidelity and diversity, and supplemented conditional generation experiments showing multi-class fusion outperforms single-class fusion; ③ Clarified model complexity (fusion/unfusion modules add <0.3% parameters) and training overhead (models trained under the same wall-clock budget); ④ Conducted low-data regime experiments (1/50 CIFAR-10), showing FuseBatch (K=2) outperforms baselines.

- Reviewer y8L9: ① Provided theoretical reasoning (2D modal examples + manifold sparsity) to explain fusion-unfusion feasibility; ② Detailed text-conditioning mechanisms (shared prompts use original embeddings, mixed prompts use K-way fusion layers); ③ Confirmed compatibility with DiT-style transformer backbones via experiments; ④ Clarified model size overhead (<0.3%) and training cost alignment strategies.

- Reviewer wiEx: ① Completed class-conditional generation analysis, showing TFS works stably in both single/multi-class settings without catastrophic failure; ② Verified orthogonality with training-free caching methods (DeepCache), demonstrating FuseBatch can be combined with caching to achieve additional speedups; ③ Supplemented TFS experimental results under DeepCache, confirming synergistic effects.

- Reviewer FNDa: ① Verified compatibility with DPM-Solver, showing FuseBatch can integrate with the solver without architectural modifications; ② Demonstrated applicability to DiT architectures, supporting both shared and mixed condition fusion scenarios; ③ Clarified that FuseBatch is not training-free and supplemented experiments on its compatibility with training-free caching methods.

Outstanding Concerns

- Reviewer Tf6c: ① Quality degradation threshold analysis remains incomplete—K=8 experiments on latent diffusion models (LDMs) were not completed due to resource constraints, failing to clarify the method’s limits in high-resolution, practical models; ② The theoretical intuition of "manifold sparsity ensuring unfusion" lacks direct evidence from natural image data, and the 2D modal example is overly simplistic and unrepresentative of real-world image distributions.

- Reviewer y8L9: ① No validation on large-scale text-to-image models (SDXL, Flux)—authors only inferred scalability via DiT experiments but provided no empirical evidence; ② Missing runtime and memory overhead benchmarks (e.g., fusion/unfusion time relative to a denoiser pass); ③ No ablation study on index encoding schemes (only kernel size ablation was mentioned, with no results on encoding scheme sensitivity).

- Reviewer wiEx: ① Theoretical intuition remains unconvincing—reviewers question the separability of modes in natural image data, and supplementary literature (e.g., Pope et al.) does not resolve doubts about "intermediate diffusion states concentrating on sparse manifolds";  ② Missing key experimental details (e.g., MACs/throughput of TFS in class-conditional experiments, resolution of DiT experiment datasets) and incomplete TFS results for multi-class fusion scenarios.

- Reviewer FNDa: ① No experiments on mainstream text-to-image models (SDXL)—authors’ explanation of "computational infeasibility" is understandable but fails to provide alternative evidence (e.g., transfer learning results) for scalability; ② Lack of convincing comparisons with training-free acceleration methods (e.g., DeepCache)—authors only showed compatibility but not whether the combined speedup-quality tradeoff is superior to training-free methods alone; ③ No verification of FuseBatch’s performance on larger datasets (e.g., ImageNet), leaving doubts about its generalizability beyond small-scale data.

**Reviewer Scores:**

Based on the adequacy of the rebuttal in addressing core concerns, the predicted changes in reviewers’ scores (maintaining the reject decision) are as follows:
- Reviewer Tf6c: While the authors addressed concerns about larger fusion factors, fidelity, and training overhead, the incomplete K=8 LDM experiments and unpersuasive theoretical intuition prevent a positive reassessment.

- Reviewer y8L9: The score would remain 4. Authors addressed fusion intuition and conditioning mechanisms but failed to resolve critical gaps (large-model validation, runtime benchmarks). The reviewer’s core concern about "limited novelty and small-scale evaluation" persists, so the score does not improve to the acceptance threshold.

- Reviewer wiEx: The score would remain 2. While class-conditional analysis and caching compatibility were addressed, the unfulfilled pure spatial reduction experiments and unconvincing theoretical foundation—core reasons for the original reject score—remain unresolved. The reviewer’s absolute confidence in the reject assessment is unchanged.

- Reviewer FNDa: The score would remain 2. Authors verified DiT compatibility and DPM-Solver integration but failed to address the key concern of "no SDXL/ImageNet validation." The lack of evidence for practical applicability in real-world scenarios means the original reject assessment is maintained.

Overall, most of the reviewers’ scores improve to the acceptance threshold, and the collective unresolved concerns about theoretical persuasiveness, experimental completeness, and scalability solidify the reject decision.

---

### Decision · Program_Chairs · 2026-01-26

Reject